# Development of a novel multi-epitope mRNA vaccine candidate to combat SFTSV pandemic

**Fei Zhu**[1,2,3,4,5☯], **Shiyang Ma**[1,2,3,4,5☯], **Yizhong Xu**[1,2,3,4,5], **Ziyou Zhou**[1,2,3,4,5], **Peipei Zhang**[1,2,3,4,5], **Wenzhong Peng**[1,2,3,4,5], **Hang Yang**[1,2,3,4,5], **Caixia Tan**[6], **Jie Chen**[1,2,3,4,5]\*, **Pinhua Pan**[iD][1,2,3,4,5]\*

**1** Department of Respiratory Medicine, National Key Clinical Specialty, Branch of National Clinical Research Center for Respiratory Disease, Xiangya Hospital, Central South University, Changsha, China, **2** Center of Respiratory Medicine, Xiangya Hospital, Central South University, Changsha, China, **3** Clinical Research Center for Respiratory Diseases in Hunan Province, Changsha, China, **4** Hunan Engineering Research Center for Intelligent Diagnosis and Treatment of Respiratory Disease, Changsha, China, **5** National Clinical Research Center for Geriatric Disorders, Xiangya Hospital, Changsha, China, **6** Department of Infection Control Center, Xiangya Hospital, Central South University, Changsha, China

☯ These authors contributed equally to this work.
\* pinhuapan668@csu.edu.cn (PP); chenjie869@csu.edu.cn (JC)

## Abstract

*Severe Fever with Thrombocytopenia Syndrome virus* (SFTSV) is a novel identified pathogen, despite two decades of research on SFTSV, the potential widespread threats pose a significant challenge for researchers in developing new treatment and prevention methods. In this present, we have developed a multi-epitope mRNA vaccine for SFTSV and valid it with in silico methods. We screened 9 immunodominant epitopes for cytotoxic T cells (CTL), 7 for helper T cells (HTL), and 8 for Linear B-cell (LBL) based on promising candidate protein Gn, Gc, Np, and NSs. All predicted epitopes demonstrated strong antigenicity without any potential harm to humans. Additionally, the high conservancy is required to cover different strains. All epitopes as well as adjuvants were constructed into a final vaccine, which was further assesd by calculating of physicochemical properties. Then, we docked the vaccine protein with immune receptors and analyzed the complexes with dynamic simulations to evaluate its affinity to receptors. Finally, the vaccine sequence was constructed into a mRNA sequence. The constructed vaccine is a potential candidate for combating SFTSV by stimulating protective humoral and cellular immune responses.

## Author summary

As a common situation, when pathogens enter the body, antigenic proteins (such as virus proteins synthesized by infected cells) will be processed into short peptides, where those immunogenic peptides will bind to MHC molecules to form complexes, triggering immune processes. These short peptides are called epitopes, and on a larger antigenic protein, only a small number of epitopes actually function. Through the linkage of epitopes, multi-epitope subunit vaccines can be formed, which exhibit excellent properties and, at the same time, reduce the potential harm to the human body compared to natural

**Data availability statement:** The data that support the findings of this study are available in the Supporting information.

**Funding:** This work is supported by the Key R&D Program of Hunan Province (No. 2022SK2038 to P. P); Natural Science Foundation of ChangSha (No. kq2208368 to P. P); Natural Science Foundation of Hunan Province of China (No. 2023JJ30930 to P. P); The Scientific Research Program of FuRong Laboratory (No. 2023SK2101 to P. P); National Natural Science Foundation of China (No. 81770080); National Natural Science Foundation of China (No. 8210012334 to P. P); Project Program of central south university graduate education teaching reform (No. 2022JGB025 to P. P); The national key clinical specialist construction programs of China (No. z047-02 to P. P); Research Project of Teaching Reform in Colleges and Universities in Hunan Province (No. 2021jy139-2 to P. P); China Postdoctoral Science Foundation (No. 2022M713520 to P. P) and Research Project on teaching reform of ordinary colleges and universities in Hunan province (No. 2022JGYB037 to P. P). The funders had no role in study design, data collection and analysis, decision to publish, or preparation of the manuscript.

**Competing interests:** The authors have declared that no competing interests exist.

protein vaccines. With the advancement of immunoinformatics technology, we can now predict unknown epitopes from proteins. Various studies have shown that this prediction has high accuracy and can be fully utilized in vaccine design. Based on this, when emerging pathogens such as SFTSV appear, we can significantly reduce the time required for vaccine design and validation.

## Introduction

Nowadays, the development of pathogen isolation and identification techniques has clarified the etiology of more diseases, and researchers have identified culprits for many diseases that lack specific symptoms. Since first reported in 2009, SFTSV has been widely distributed through East Asia [1–3]. It is emerging as an arthropod-borne infectious disease, causing a significant number of new cases every year, during the period of 2013 to 2016, 7419 cases in total were reported across China, among which 355 cases have been confirmed to be fatal [4]. Due to a lack of awareness, this data is underestimated, a large number of patients who have not undergone high-throughput sequencing are likely to be perceived as atypical cases of hemolytic uremic syndrome. Meanwhile, the lack of an adequate number of individual cases in each report leads to fluctuations in the estimated mortality rate, which varies between 2–48% in different studies. However, as the reports increase, the true mortality rate of *SFTSV* probably lies at 5%–10% and tends to increase significantly with age [5]. Just as the name of the virus describes, *SFTSV* lead to severe fever and thrombocytopenia syndrome, which usually presents with Liver function impairment and gastrointestinal symptoms, cytokine storms, and disseminated intravascular coagulation (DIC) caused by viral attack finally leads to multiple organ dysfunction [6]. *SFTSV* was first recognized as a tick-borne virus, as early reports all have a clear history of tick bites. However, subsequent reports indicate that *SFTSV* can be transmitted through fluid contact or even aerosols, and some cases of cluster infections within households or hospitals have emerged [7]. Simultaneously, cases of direct transmission of *STFSV* from animals to humans without tick vectors have also been reported [8]. In such an unsettling atmosphere, WHO ranked STFS the highest research priority in 2017, due to the potential risk of a pandemic. Despite the pathogenic process of *SFTSV* gradually become clear, how to treat and prevent it remains a challenge. Considering there is a general lack of specific immunity in the population, vaccine development is an important component in the prevention and control of *SFTSV*.

According to phylogenetic analysis based on gene sequencing results, *STFSV* was classified into the *Bunyavirales Phenuiviridae* family, and assigned as species *Dabie bandavirus* in genus *Bandavirus*, precisely. Like other Bunyaviruses, *SFTSV* is a spherical virus packaging a single negative-stranded RNA, three mRNA segments were named L, M, and S based on their relative length [9]. L segment encodes RNA-dependent RNA polymerase (RdRp), and participates in viral genetic informations' transmission. While S encodes nucleocapsid protein (NP) as well as nonstructural protein (NSs). Finally, as the segment with the highest level of self-replicating, M segment encodes the precursor of membrane protein (Gp), which will be modified into glycoprotein c(Gc) and glycoprotein n(Gn) later. In the proteome of *STFSV*, NP, NSs, Gc, and Gn are highly associated with the immune progress of *STFSV*. Gc and Gn form protein polymers on the envelope of *STFSV* and mediates virus invasion into target cells [10]. Considering the significant importance of Gc and Gn in viral life and the characteristics of providing main neutralizing epitopes and immunogenicity, they are currently the main target for antiviral intervention. Studies have revealed antibody titers of Gn and Gc are associated

closely with patient survival [11]. Antibodies obtained from camels immunized with Gn can suppress *SFTSV* infection in mice [12]. Vesicular stomatitis virus (VSV) vector-based Gc/Gn vaccine can induce strong neutralizing antibodies and protect mice from fatal *SFTSV* attack [13], similarly, another smallpox (LC16m8) vector-based Gc vaccine performed well in surviving mice and ferret models from fatal *SFTSV* infection [14]. Meanwhile, Kwak et al developed a series of DNA vaccines encoding Ns, NP, RdRp, Gc, and Gn separately, among which, only the Gc and Gn vaccine can generate neutralizing antibodies, but non-glycoproteins (Ns, Np, and RdRp) can still provide protection for ferret from fatal *SFTSV* infection, which is mainly achieved through the induction of specific T cell immunity [15]. Though there was a study that reported recombinant Ns has no effect on the clearance of *SFTSV* in mice [16], in doctor Kwak's research, individual Ns DNA vaccine can provide partial protection, NP and RdRp also have the same effect [15]. Overall, the current research on *SFTSV* vaccines is still somewhat insufficient.

MRNA is an ideal vaccine platform because of its unique advantages in improving efficacy, reducing development time, ensuring safety profile, and minimizing manufacturing cost [17]. MRNA vaccines carrying multiple epitopes can lead to activation of both humoral and cellular immune response. At this present, our team predicted conservative T and B cell epitopes from protective target proteins Gn, Gc, NP, and Ns, which are critical in virus invasion and replication. After sorting and selecting, an epitope-based mRNA vaccine was designed in the end, further prediction of the vaccine docking to immune receptors and dynamic validated the potential of the construction. Hopefully, this candidate vaccine can address potential pandemics of *SFTSV*.

## Methods

### 1. Target protein sequence acquisition

Gn, Gc, NP, and NSs of the SFTSV reference strain (GCA_003087855.1), HBMC strain, and HB29 strain were retrieved from the NCBI database, since HBMC and HB29 strains are common in China (https://www.ncbi.nlm.nih.gov/).

### 2. Prediction of epitopes

**2.1. Prediction of CTL epitopes.** NetCTL 1.2 (https://services.healthtech.dtu.dk/services/NetCTL-1.2/) predicted CTL epitopes with a threshold of 0.75. All 12 supertypes (A1, A2, A3, A24, A26, B7, B8, B27, B39, B44, B58, B62) were selected for epitope prediction, peptide-MHCI binding, C-terminal cleavage of the proteasome and、 TAP transport efficiency were considered in this step [18]. Then, Vaxijen 2.0 predicted the antigenicity of epitopes, only epitopes with antigenicity >0.4 will be included in the next screening step [19]. AllerTOP 2.0 (http://www.ddg-pharmfac.net/AllerTOP) and ToxinPred 2.0 (https://webs.iiitd.edu.in/raghava/toxinpred2/) were used to predict the allergenicity and toxicity of the epitopes, only epitopes that are non-toxic and non-allergenic will be further screened [20,21]. To ensure the CTL epitopes has proper immunogenicity, the MHC I Immunogenicity tool from IEDB (http://tools.iedb.org/immunogenicity/) was used to predict the immunogenicity of the epitopes, only epitopes with an immunogenicity >0 will be selected [22]. Finally, the Tepitool tool from IEDB (http://tools.iedb.org/tepitool/) used Consensus algorithm to predict the MHC I alleles that the epitopes can bind to, only epitopes with an IC50 < 500 nm will be included in the vaccine construction [23].

**2.2. Prediction of HTL epitopes.** The NetMHCIIpan4.0 server (https://services.healthtech.dtu.dk/services/NetMHCIIpan-4.0/) was utilized for the prediction of HTL epitopes, by deploying NNAlign_MA algorithm, NetMHCIIpan can predict HTL epitopes

base on combined methods of prediction algorithm training and the capability of annotating EL (eluted ligands) MA (Multi Allelic) sequences to single MHC restrictions [24]. A total of 26 commonly observed MHC II alleles (DRB1_0101, DRB1_0301, DRB1_0401, DRB1_0404, DRB1_0405, DRB1_0701, DRB1_0802, DRB1_0901, DRB1_1101, DRB1_1302, DRB1_1501, DRB3_0101, DRB4_0101, DRB5_0101, HLA-DQA10501-DQB10201, HLA-DQA10501-DQB10301, HLA-DQA10301-DQB10302, HLA-DQA10401-DQB10402, HLA-DQA10101-DQB10501, HLA-DQA10102-DQB10602, HLA-DPA10201-DPB10101, HLA-DPA10103-DPB10201, HLA-DPA10103-DPB10401, HLA-DPA10301-DPB10402, HLA-DPA10201-DPB10501, HLA-DPA10201-DPB11401) were chosen for the prediction of their corresponding epitopes. Epitopes possessing an IC50 value less than 500 nm were incorporated in the subsequent screening phase. Vaxijen2.0, Allertop 2.0, as well as ToxinPred 2.0 were employed to forecast epitopes' antigenicity, allergenicity, and toxicity, respectively. Only epitopes displaying an antigenicity score above 0.4 while lacking toxicity or allergenicity could be introduced into the next step. IFNepitope (https://webs.iiitd.edu.in/raghava/ifnepitope/), IL2Pred (https://webs.iiitd.edu.in/raghava/il2pred/), and IL4Pred (https://webs.iiitd.edu.in/raghava/il4pred/) determined the potential of the epitopes to induce IFN-γ, IL-2, and IL-4, correspondingly [25,26]. The potential ability of inducing pro-inflammatory factors is important in HTL epitopes mediating the activation, proliferation, and differentiation of T and B cells [27]. Subsequently, only epitopes that can stimulate the production of all three cytokines were selected.

**2.3. Prediction of linear B cell epitopes.** The ABCpred tool (https://webs.iiitd.edu.in/raghava/abcpred/) was employed to predict linear B cell epitopes, utilizing a threshold value of 0.51, this algorithm is mainly based on standard feed-forward (FNN) and recurrent neural network (RNN) [28]. The predicted length was determined to be 16 amino acids. Vaxijen2.0, Allertop 2.0, and ToxinPred 2.0 were checked the antigenicity, allergenicity, and toxicity. Subsequently, only epitopes exhibiting an antigenicity score exceeding 0.4, while remaining devoid of any toxicity or allergenicity, were selected for further screening.

## 3. Epitope conservation prediction and homonology analyzation

A total of 230 complete genomes of the *SFTSV* were downloaded from the NCBI database on November 20, 2022, which was used for the purpose of constructing a local comprehensive genome database. The local tblastn tool was proceeded to forecast the conservation of epitopes across the 230 *SFTSV* virus strains. Only epitopes exhibiting significant conservation or free from mutation sites were incorporated into the final vaccine assembly. Meanwhile, another blast was ran to ensure that there is no overlap between the epitopes and the human genome and the human gut microbiome genome to avoid potential harm. We downloaded the complete proteome of humans (Txid:9606) from the NCBI database as well as the complete proteome of 79 common intestinal bacterial species in the human body from the NCBI database [29]. Use the Diamond software to construct a local database and analyze the homology of epitopes with the human and intestinal bacterial proteomes using the BLASTp program. Epitopes with an E-value less than 0.00001 and a bit-score >100 are considered homologous to the human and intestinal bacterial proteomes.

## 4. Construction of vaccines and prediction of their physicochemical properties

Ultimately, we obtained epitopes who has strong antigenicity and without toxicity or allergenicity to construct the vaccine. CTL epitopes are connected with AAY linkers, HTL epitopes are connected using the GPGPG linkers, and B-cell epitopes are connected with KK linkers.

The β-defensin ll is attached to the N-termini of the vaccine along with the use of an EAAAK linker, who serves as an adjuvant enhancing the immunogenicity. And a TAT sequence is attached to the C-terminal with an EAAAK linker to assist the vaccine transfer across the cellular membrane. The Protparam tool (https://web.expasy.org/protparam/) was utilized to predict the physicochemical properties of vaccines. Then, ToxinPred2.0 and AllerTOP 2.0 were employed to predict the toxicity and allergenicity of vaccines, while Vaixijen 2.0 and ANTIGENpro (http://scratch.proteomics.ics.uci.edu/) were employed for the prediction of vaccine antigenicity [30].

## 5. Prediction of secondary and tertiary structures of the vaccine

PISPRED 4.0 (http://bioinf.cs.ucl.ac.uk/psipred/) was employed to predict the secondary structure of vaccines with an accuracy rate exceeding 84%. The Robetta (https://robetta.bak-erlab.org/) software was utilized for predicting the tertiary structure of vaccines. This software employs the RoseTTAfold method to predict the tertiary structure of proteins, yielding results that approach the accuracy of Deepmind [31].

## 6. Refinement of tertiary structure and validation of model quality

To improve the overall quality of the model, the GalaxyRefine tool (https://www.bio.tools/galaxyrefine#!) refined and optimized the protein structure, selecting models with lower Mol-probity scores for subsequent analysis [32]. Finally, the overall quality of the protein structure is validated using PROCHECK, ERRAT, and Prosa-Web tools (https://saves.mbi.ucla.edu/) [33,34]. The visualization of vaccine structures is performed using PyMOL v2.5 software and Chimera v1.15 software.

## 7. Prediction of conformational B cell epitopes

The antibody epitope prediction tool ElliPro (http://tools.iedb.org/ellipro/) was utilized to predict potential conformational B cell epitopes in vaccine structures. ElliPro predicts discontinuous antibody epitopes by leveraging the three-dimensional structure of the antigenic protein and associates each predicted epitope with a corresponding protrusion index (PI) value [35].

## 8. Molecular docking

Toll-like receptor 2 (TLR2) and Toll-like receptor 4 (TLR4) are essential for recognizing viral membrane proteins and initiating the innate immune response in the host. The HLA-A0201 and HLA-DRB1_0101 receptors are common MHC molecule receptors that are crucial for antigen presentation. The crystal structures of TLR2 (PDB ID: 2Z7X), TLR4 (PDB ID: 4G8A), HLA-A0201 (PDB ID: 4U6Y), and HLA-DRB1_0101 (PDB ID: 5V4N) were retrieved from the RCSB PDB database. The ClusPro 2.0 server (https://cluspro.org/help.php), a highly regarded server for predicting protein binding, was utilized for molecular docking of these molecules [36]. Then, the HADDOCK 2.4 server (https://wenmr.science.uu.nl/haddock2.4/) was employed to refine and optimize the structures of protein complexes in a rational manner [37]. Finally, the PDBsum server (https://www.ebi.ac.uk/thornton-srv/databases/pdbsum/) was utilized to analyze interactions between vaccines and receptors [38].

## 9. Molecular dynamics simulation

Molecular dynamics simulations proceeded to model the stability of vaccine-receptor complexes in aqueous solutions, thereby elucidating the binding stability between vaccines and receptors. The Gromacs v2022.1 software was employed to conduct 100 ns molecular

dynamics simulations on all vaccine-receptor complexes with the Amber14SB force field. And the vaccine-receptor complexes were confined within a cubic box containing TIP3P water molecules. Na+ or Cl- ions were subsequently introduced to achieve overall system charge neutralization. Then, energy minimization was conducted to attain the protein complexes' lowest energy state. The entire system was equilibrated at a temperature of 310 K (close to body temperature) through NVT (canonical ensemble) simulations for 200 ps. Following that, NPT (isothermal-isobaric ensemble) simulations were performed on the system at a pressure of 1 atm for 1 ns. Finally, the protein complex was subjected to a 100 ns molecular dynamics simulation in an aqueous solution at 310 K and 1 atm using Gromacs. RMSD (root mean square deviation), RMSF (root mean square fluctuation), the number of hydrogen bonds, and Rg (gyration radius) were assessed based on the molecular dynamic simulation trajectory using Gromacs [39].

## 10. MMPBSA (Molecular Mechanics/ Poisson Boltzmann Surface Area) energy analysis

GMX_MMPBSA v1.56 was utilized for calculating the changes in binding energy between protein complexes in simulated trajectories. The MMPBSA method is used to assess the binding energies among protein molecules during the last 20 ns of the simulation, assuming the protein complex is in a stable state. The specific formula employed for the calculation is as follow:

$$\Delta G^0_{0bind,\,solv} = \Delta G^0_{0bind,\,solv} + \Delta G^0_{solv,\,complex} - \left( \Delta G^0_{solv,\,ligand} + \Delta G^0_{solv,\,receptor} \right) \tag{1}$$

$$\Delta G^0_{solv} = \Delta G^0_{electrostatic,\,c=80} - \Delta G^0_{electrostatic,\,c=1} + \Delta G^0_{hydrophobic} \tag{2}$$

$$\Delta G^0_{vaccum} = \Delta E^0_{molecular\,mechanics} - T \cdot \Delta S^0_{normal\,mode\,analysis} \tag{3}$$

## 11. Population coverage and immune simulation

The frequencies of HLA alleles vary across different regions and populations. Thereby, the Population coverage tool of IEDB (http://tools.iedb.org/population/) calculates the global coverage of the vaccine based on the specific HLA molecules that correspond with various epitopes [40].

The C-IMMSIM server (https://kraken.iac.rm.cnr.it/C-IMMSIM/index.php?page=0) simulates the host's immune response following three vaccine injections with a 4-week interval between each. By default, the HLA-A01:01, HLA-B07:02, and HLA-DRB1*01:01 allele gene molecules are chosen. Overall, the simulation lasts for a total of 350 days [41].

## 12. Construction of multi-epitope mRNA vaccine

Jcat server (http://www.jcat.de/Start.jsp.) was utilized for reverse translation and optimization of the vaccine. For the construction of the definitive multi-epitope mRNA vaccine, the DNA sequence of the vaccine was modified by adding a Kozak sequence at the 5' end to enhance RNA stability and translation efficiency [42], signal peptide Tissue Plasminogen Activator (tPA) follows the Kozak sequence to facilitate the extracellular secretion of proteins and improve antigen presentation [43]. Additionally, an MHC I-targeting domain (MITD) sequence is added to the 3' end of the vaccine DNA sequence to facilitate the presentation of CTL epitopes [44], then a TAA condon ends the translation. Besides, to stabilize the mRNA

partial sequence of cytomegalovirus immediate-early gene Untranslated Region (UTR) was added as 5' untranslated region (UTR), and a partial sequence of the human growth hormone was attached as 3' UTR [45,46]. The full DNA sequence was translated into mRNA using the Transcription Tool. the RNAfold server (http://rna.tbi.univie.ac.at/cgi-bin/RNAWebSuite/RNAfold.cgi) and mFold server (http://www.bioinfo.rpi.edu/applications/mfold) predicted the mRNA's secondary structure.

TLR 3, 7, 8, and 9 have a high affinity for nucleic acid-related pathogen-associated molecular patterns (PAMPs), especially the ssRNA genome of viruses, which are referred to as virus-associated molecular patterns (VAMPs) in this context. Therefore, we examined the interaction between mRNA vaccine sequences and several TLR molecules [47]. The sequences of TLR3, TLR7, TLR8, and TLR9 were obtained from the Uniprot database. The retrieved RNA sequences were processed using the imRNA tool developed by the Indraprastha Institute of Information Technology in New Delhi (https://webs.iiitd.edu.in/raghava/imrna/) [48]. This tool is based on the Motif-EmeRging and Classes-Identification (MERCI) program, which can scan RNA sequences to identify potential motifs with immunomodulatory properties. The RPISeq tool from Iowa State University (http://pridb.gdcb.iastate.edu/RPISeq/) provides a cost-effective method to predict RNA-protein interactions by analyzing RNA and protein sequence data. The RPISeq tool runs on machine learning algorithms and generates output for RNA-protein interaction predictions in two main forms - Random Forest (RF) classifier and Support Vector Machine (SVM) classifier [49]. Further docking analysis was conducted on TLR molecules that can bind to mRNA vaccines. The interaction between vaccine mRNA and the TLR receptor was predicted using the HDOCK server. Subsequently, PyMOL was used for further visualization of the interaction between TLR receptor and mRNA.

## 13. Vector construction

The Jcat server is used for reverse translation and codon optimization of vaccine sequences. Escherichia coli K12 was chosen as the host to avoid optimization of the cDNA sequence of the vaccine for the XhoI and BamHI restriction enzyme sites. The cDNA sequence was inserted into the pet28a (+) vector using SnapGene software.

## Results

### 1. Prediction of epitopes

Through screening with diverse epitope prediction tools, a total of 9 CTL epitopes (Table 1), 7 HTL epitopes (Table 2), and 8 linear B-cell epitopes (Table 3) were obtained. All epitopes exhibit antigenicity without toxicity or allergenicity, and the 7 HTL epitopes can stimulate the production of IFN-γ, IL-2, and IL-4.

By comparing the constructed SFTS virus genome library, it was found that most epitopes are conserved in a minimum of 170 strains, except for the epitope KSTEIHFHSGSLVGK, which is conserved in only 25 strains. Meanwhile, the combination of KSTEIQFHSGSLVGK and KSTEIHFHSGSLVGK epitopes provides coverage for nearly all virus strains in the genome library.

### 2. Construction of the vaccine and prediction of its physicochemical properties

The CTL, HTL, and B-cell epitopes are connected with AAY, GPGPG, and KK linkers, finally forming a vaccine protein comprising 486 amino acids (Fig 1A). The vaccine's antigenicity

**Table 1. Screening of CTL epitopes.**

| Protein | Epitope | Start position | HLA molecule | Rank% | IC50 | Anrigenicity | Conservation |
|---|---|---|---|---|---|---|---|
| Gc | SSYFVPDAR | 71 | HLA-A*11:01 | 0.333 | 54.84 | 0.8414 | 221/230 |
| | | | HLA-A*31:01 | 0.276 | 31.2 | | |
| | | | HLA-A*33:01 | 0.662 | 315.1 | | |
| | | | HLA-A*68:01 | 0.081 | 8.04 | | |
| | TSVEAVANY | 284 | HLA-A*26:01 | 0.038 | 69.15 | 0.4936 | 222/230 |
| | | | HLA-A*30:02 | 0.476 | 163.58 | | |
| | | | HLA-B*15:01 | 0.902 | 186.67 | | |
| | | | HLA-B*35:01 | 0.362 | 161.26 | | |
| | KQVFRSRTK | 501 | HLA-A*03:01 | 0.639 | 200.46 | 0.4441 | 198/230 |
| | | | HLA-A*30:01 | 0.17 | 31.22 | | |
| | | | HLA-A*31:01 | 1.541 | 252.5 | | |
| | VSLSFDHAV | 399 | HLA-A*02:06 | 1.317 | 129.95 | 1.1072 | 226/230 |
| Gn | PTFDGYVGW | 114 | HLA-B*57:01 | 0.524 | 271.96 | 1.3178 | 229/230 |
| | | | HLA-B*58:01 | 0.744 | 189.04 | | |
| NSs | KPSVWFLQA | 105 | HLA-B*07:02 | 0.807 | 464.85 | 0.5315 | 229/230 |
| Np | YLPVGPAVM | 125 | HLA-A*02:03 | 2.586 | 187.28 | 1.1170 | 229/230 |
| | MAFGSLIPT | 147 | HLA-A*68:02 | 0.148 | 18.24 | 0.7402 | 230/230 |
| | FTKTINVKM | 177 | HLA-A*68:02 | 1.203 | 230.48 | 0.4427 | 230/230 |

**Table 2. Screening of HTL epitopes.**

| Protein | Epitope | Start Position | HLA II molecule | Antigenicity | Toxin | Allergic | IFN-γ | IL-2 | IL-4 | Conservation |
|---|---|---|---|---|---|---|---|---|---|---|
| Gc | AAWVPSAVIELTMPS | 164 | DRB1_0701 | 0.4615 | – | – | + | + | + | 227/230 |
| | | | DRB1_0901 | | | | | | | |
| | | | HLA-DQA10501-DQB10201 | | | | | | | |
| Gn | QPFDVAWMDVGHSHK | 204 | DRB4_0101 | 1.5028 | – | – | + | + | + | 219/230 |
| | | | HLA-DQA10401-DQB10402 | | | | | | | |
| | KSTEIHFHSGSLVGK | 370 | DRB1_0101 | 1.3039 | – | – | + | + | + | 25/230 |
| | | | DRB1_0701 | | | | | | | |
| | | | DRB1_0901 | | | | | | | |
| | | | DRB1_1302 | | | | | | | |
| | | | DRB1_1501 | | | | | | | |
| | KSTEIQFHSGSLVGK | 370 | DRB1_0101 | 1.1427 | – | – | + | + | + | 199/230 |
| | | | DRB1_0701 | | | | | | | |
| | | | DRB1_0901 | | | | | | | |
| | | | DRB1_1302 | | | | | | | |
| | | | DRB1_1501 | | | | | | | |
| | HSQFQGYVGQRGGRS | 45 | DRB1_0101 | 0.9471 | – | – | + | + | + | 224/230 |
| | | | DRB1_0401 | | | | | | | |
| | | | DRB1_0901 | | | | | | | |
| | | | DRB1_1501 | | | | | | | |
| | | | DRB5_0101 | | | | | | | |
| | | | HLA-DQA10401-DQB10402 | | | | | | | |
| | TFLELKSFSQSEFPD | 171 | HLA-DPA10103-DPB10201 | 0.75 | – | – | + | + | + | 223/230 |
| Np | LMALQEKYGLVERAE | 83 | HLA-DQA10501-DQB10301 | 0.9759 | – | – | + | + | + | 208/230 |

**Table 3. Screening of LBL epitopes.**

| Protein | Epitope | Position | Antigenicity | Conservation |
|---------|---------|----------|--------------|--------------|
| Gc | IRGSFSVNYRGLRLSL | 324 | 0.9948 | 219 |
| | SVTYLGSDMEVSGLTD | 199 | 0.4814 | 223 |
| Gn | IIVILLGYAGLMLLTN | 438 | 0.5989 | 214 |
| | SQVSYYPAENSYSRWS | 59 | 0.5439 | 178 |
| Np | GSKRLMALQEKYGLVE | 79 | 0.5269 | 230 |
| NSs | FFSIKNSWAMETGREN | 117 | 0.9532 | 223 |
| | VWFLQAAHMFFSIKNS | 108 | 0.8815 | 228 |
| | MNANTVRLEPSLGEYP | 16 | 0.5888 | 209 |

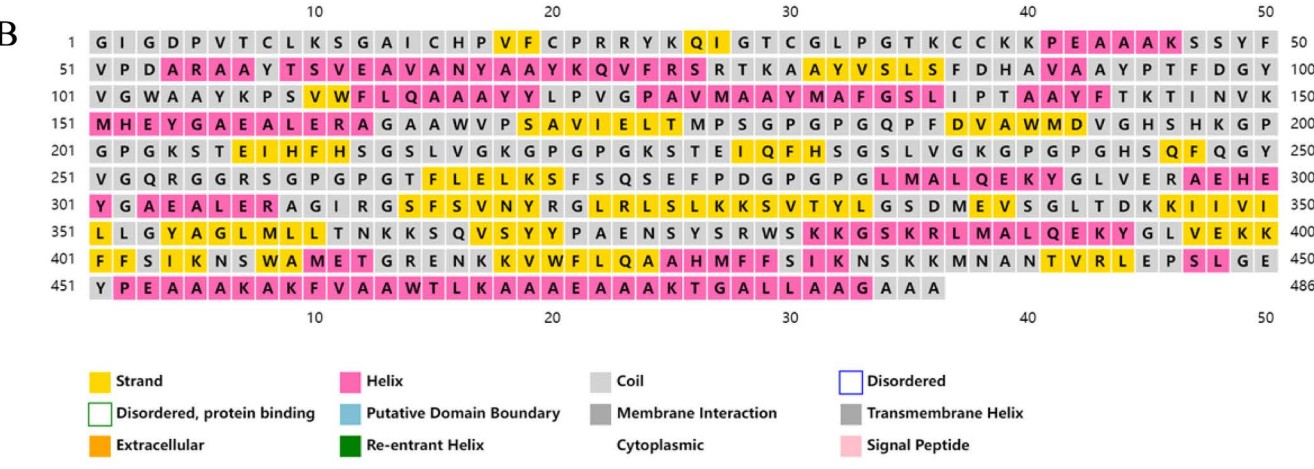

**Fig 1. Full sequence and secondary structure of vaccine construction.** (A) The full-length amino acid sequence of the vaccine; Green represents the β-defensin ‖ adjuvant, blue represents the CTL epitope, red represents the HTL epitope, orange represents the linear B cell epitope, purple represents the pan-HTL epitope, and yellow represents the TAT peptide; (B) The secondary structure of the vaccine.

was calculated to be 0.6948 and 0.856963 by Vaxijen 2.0 and ANTIGENpro, indicating strong antigenicity. Meanwhile, the vaccine showed no allergenicity by when tested by AllerTOP 2.0 and AllerFP servers, and no toxicity by the ToxinPred 2.0. According to the ProtParam software, the constructed vaccine has a theoretical isoelectric point of 9.53 with a molecular weight of 52.08 kDa, it has a half-life of over 30 hours in mammalian red blood cells (in vitro), over 20 hours in yeast cells, and more than 10 hours in Escherichia coli cells. With an instability index of 33.61 and an aliphatic of 70.39, the vaccine protein is considered stable and possesses high thermal stability. With a hydrophilicity index of −0.184, the vaccine demonstrates

its hydrophilic properties. When overexpressed in Escherichia coli, SOLpro predicts a solubility probability of 0.5000 for the vaccine.

### 3. Secondary and tertiary structure prediction and validation

The vaccine's secondary structure predictions were calculated by the PISPRED, which indicated the sequence comprised 49.8% random coil, 29.6% alpha helix, and 20.6% extended strand (Fig 1B). Then, the Robetta software predicted and generated the vaccine's tertiary structure (Fig 2A), which was subsequently refined with GalaxyRefine to optimize the vaccine's side chains (Fig 2B). The vaccine achieved a Molprobity score of 1.831, indicating good model quality. The Procheck software's Ramachandran plot demonstrated that 88.8% of the amino acids resided in the preferred region, with 10.0% allowed, 0.2% generously allowed, and 1.0% disallowed (Fig 3B). The Prosa generated a Z-score of −7.64, indicating the model has a good stereochemical quality (Fig 3C). The ERRAT predicted an adequate score of 90.063 for the vaccine's tertiary structure (Fig 3D).

### 4. Conformational B cell epitopes prediction

After the screening of the vaccine's tertiary structure, ElliPro reported 7 conformational B cell epitopes in the structure (Table 4), the overall score varied from 0.528–0.855, their location in the vaccine structure are shown in Fig 4.

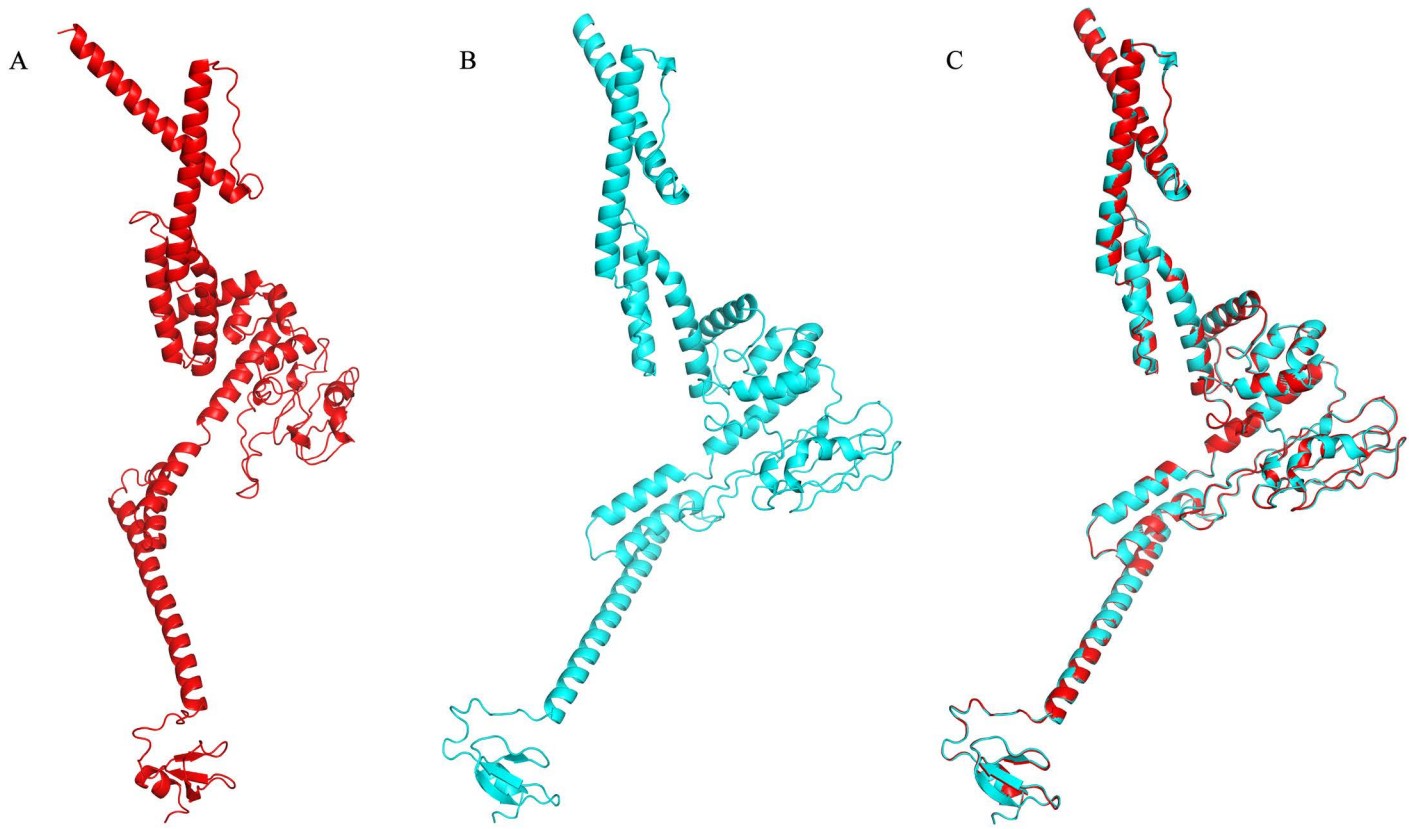

**Fig 2. Refined structure of the vaccine.** (A) Initial structure of the vaccine; (B) Refined structure of the vaccine; (C) Overlayed model of the initial structure and refined structure.

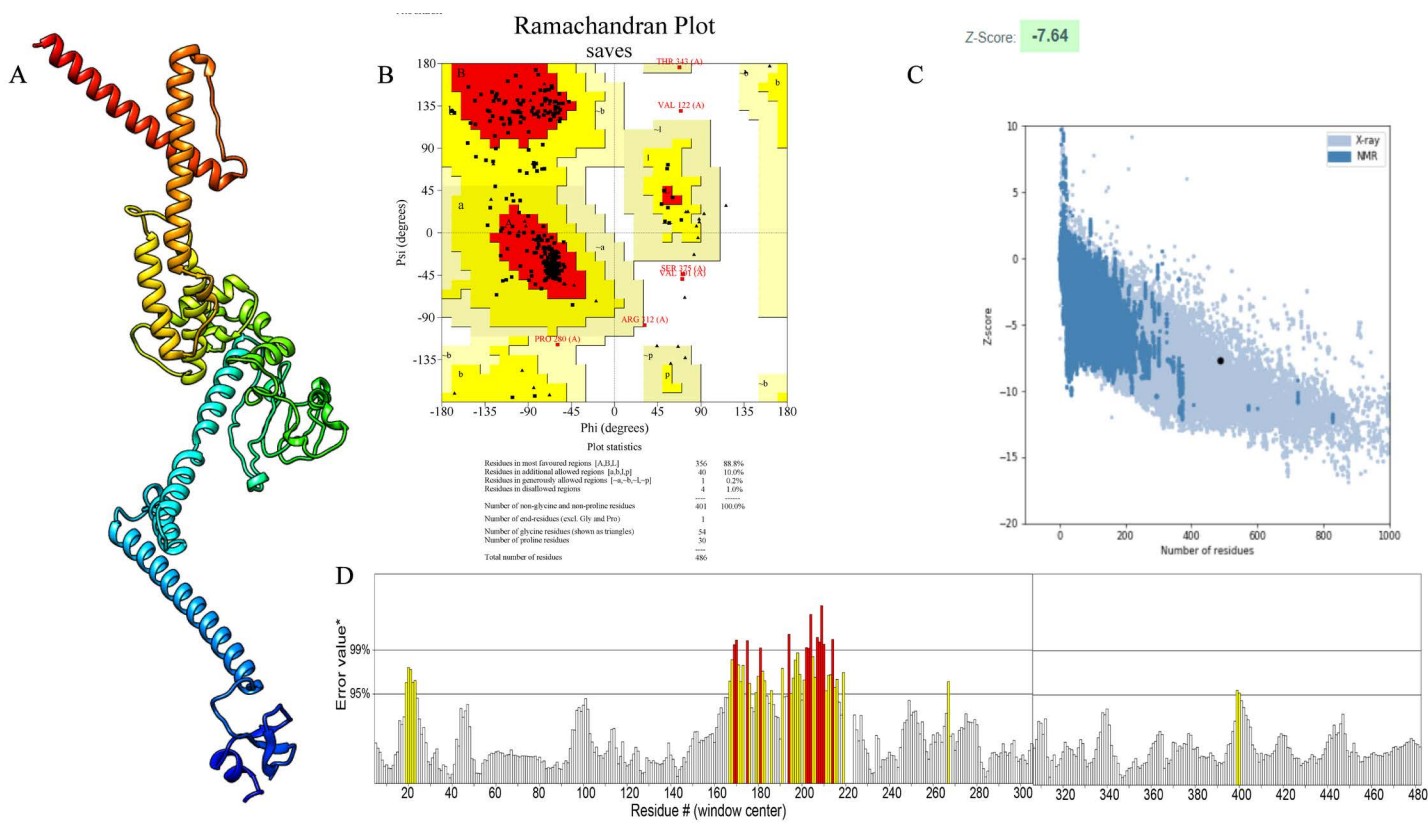

**Fig 3. Construction of multi-epitope vaccine.** (A) Visualization of the vaccine's refined structure; (B) Ramachandran plot o; (C) Z-score plot; (D) ERRAT score plot.

**Table 4. Screening of conformation B epitopes.**

| No. | Residues | Number of residues | Score |
|---|---|---|---|
| 1 | A:G1, A:I2, A:G3, A:D4, A:P5, A:V6, A:T7, A:C8, A:L9, A:K10, A:S11, A:G12, A:A13, A:I14, A:C15, A:H16, A:P17, A:V18, A:F19, A:C20, A:P21, A:R22, A:R23, A:Y24, A:Q26, A:I27, A:G28, A:T29, A:C30, A:G31, A:L32, A:P33, A:G34, A:T35, A:K36, A:C37, A:C38, A:K39, A:K40, A:P41, A:E42, A:A43, A:A44, A:A45, A:K46, A:S47, A:S48, A:Y49, A:F50, A:V51, A:P52, A:D53, A:A54, A:R55, A:A56, A:A57, A:Y58, A:T59, A:S60, A:V61, A:E62, A:A63, A:V64, A:A65, A:N66, A:Y67, A:A68, A:Y70, A:K71 | 69 | 0.855 |
| 2 | A:V367, A:S368, A:Y369, A:Y370, A:P371, A:A372, A:E373, A:N374, A:K405, A:S407, A:W408, A:M410, A:E411, A:T412, A:G413, A:R414, A:E415, A:N416, A:K417, A:K418, A:V419, A:W420, A:F421, A:L422, A:Q423, A:A424, A:A425, A:H426, A:M427, A:F428, A:F429, A:S430, A:I431, A:K432, A:N433, A:S434, A:K435, A:K436, A:M437, A:N438, A:A439, A:N440, A:T441, A:V442, A:R443, A:L444, A:E445, A:P446, A:S447, A:L448, A:G449, A:E450, A:Y451, A:P452, A:E453, A:A454, A:A455, A:A456, A:K457, A:A458, A:K459, A:F460, A:V461, A:A462, A:A463, A:W464, A:T465, A:L466, A:K467, A:A468, A:A469, A:A470, A:E471, A:A472, A:A473, A:A474, A:K475, A:T476, A:G477, A:A478, A:L479, A:L480, A:A481, A:A482, A:G483, A:A484, A:A485, A:A486 | 88 | 0.742 |
| 3 | A:A90, A:V91, A:A92, A:A93, A:Y94, A:P95, A:T96, A:F97, A:D98, A:G99, A:Y100, A:V101, A:G102, A:W103 | 14 | 0.654 |
| 5 | A:F74, A:R77, A:T78, A:A81, A:Y82, A:S84, A:L85, A:S86, A:F87, A:D88, A:H89 | 11 | 0.558 |
| 6 | A:G255, A:G256, A:R257 | 3 | 0.54 |
| 7 | A:W166, A:V167, A:P168, A:S169 | 4 | 0.528 |

## 5. Molecular docking

The vaccine was docked with HLA-DRB1_0101, HLA-A0201, TLR2, and TLR4 receptors using the ClusPro 2.0 server. We selected the docking results with higher clustering ranks, lower docking energy scores, and reasonable docking poses. To further refine these selected

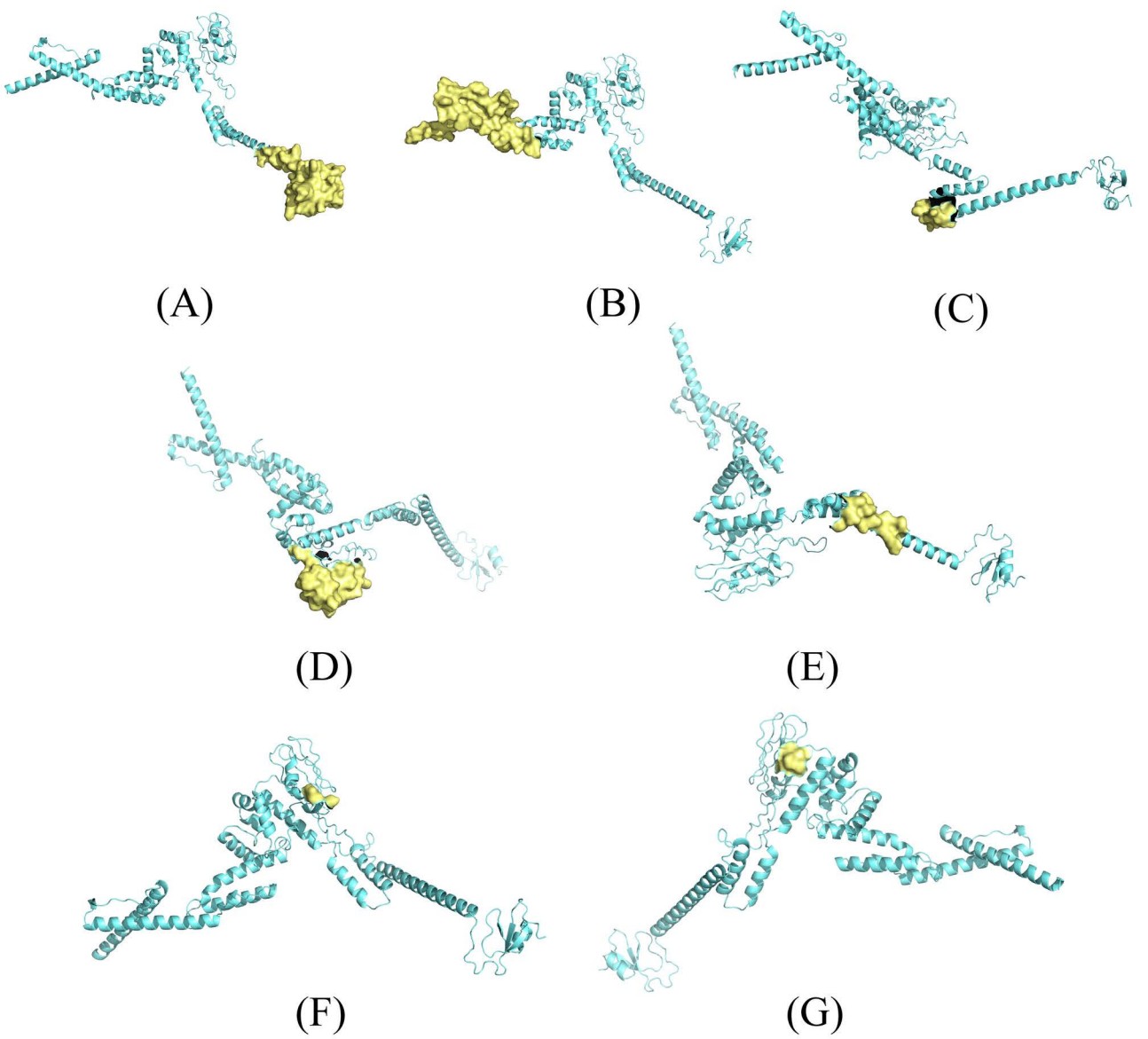

**Fig 4. Conformational epitopes screened from the constructed vaccine.** (A–G) Conformational epitope 1–7 (According to Table 4).

results, we utilized the HADDOCK 2.4 server to generate the docked complexes. Specific results can be found in Table 5. The results with low HADDOCK scores demonstrate successful interaction between the vaccine and the receptors. We employed the PDBsum software to analyze the docked complexes, aiming to elucidate the interactions between the vaccine and the receptors. The analysis revealed that the vaccine can form 18 hydrogen bonds and 5 salt bridges with HLA-A02:01 (Fig 5A), 7 salt bridges and 16 hydrogen bonds with HLA-DRB101:01 (Fig 5B), 2 salt bridges and 19 hydrogen bonds with TLR2 (Fig 6A), and 6 salt bridges and 16 hydrogen bonds with TLR4 (Fig 6B). These results suggest that the vaccine

**Table 5. Results of molecular docking.**

| Complex | Vaccine-HLA-A*02:01 | Vaccine-HLA-DRB1*01:01 | Vaccine-TLR2 | Vaccine-TLR4 |
|---|---|---|---|---|
| | ClusPro 2.0 | | | |
| Center Weighted Score | −840.7 | −1089.9 | −938.4 | −948.1 |
| Lowest Energy Weighted Score | −997.2 | −1130.9 | −1004.9 | −1119.7 |
| | HADDOCK 2.4 | | | |
| HADDOCK score | −269.7 +/− 1.6 | −698.1 +/− 2.6 | −260.9 +/− 6.0 | −276.0 +/− 3.7 |
| RMSD from the overall lowest-energy structure | 0.6 +/− 0.4 | 0.6 +/− 0.4 | 0.6 +/− 0.4 | 0.6 +/− 0.3 |
| Van der Waals energy | −136.2 +/− 3.0 | −316.0 +/− 1.4 | −145.8 +/− 3.0 | −113.4 +/− 5.2 |
| Electrostatic energy | −441.9 +/− 25.3 | −1301.2 +/− 15.7 | −324.8 +/− 16.4 | −570.8 +/− 25.5 |
| Desolvation energy | −45.1 +/− 4.7 | −121.9 +/− 4.4 | −50.1 +/− 4.1 | −48.4 +/− 3.5 |
| Buried Surface Area | 3816.4 +/− 71.2 | 8413.2 +/− 52.5 | 3631.4 +/− 29.8 | 4065.6 +/− 75.6 |
| Z-Score | 0.0 | 0.0 | 0.0 | 0.0 |
| | PDBsum | | | |
| Number of hydrogen bonds | 18 | 16 | 19 | 16 |
| Number of salt bridges | 5 | 7 | 2 | 6 |

exhibits stable binding affinity with HLA-A02:01, HLA-DRB101:01, TLR2, and TLR4 molecules, thereby eliciting host immune responses.

## 6. Molecular dynamic simulations

In order to comprehend the stability of vaccine-receptor complexes within the human body, we conducted a 100 ns molecular dynamics simulation using Gromacs v2022.1 software. RMSD (Root Mean Square Deviation) measures the difference between the simulated molecular structure and the experimental structure. The smaller the RMSD, the closer the simulation result is to the experimental result. The RMSD plot is an indicator of the structural fluctuations of the vaccine-TLRs complexes. Fig 7F illustrates that the vaccine-HLA-DRB101:01, vaccine-HLA-A02:01, vaccine-TLR2 and TLR4 complexes attained equilibrium at approximately 20 ns. The average RMSD are 1.69 (vaccine-TLR2), 1.55 (vaccine-TLR4), 1.77 (vaccine- HLA-A02:01) and 1.81 (vaccine- HLA-DRB101:01), all complexes tend to maintain a stable binding with their RMSD values fluctuating within a defined range (Fig 7A). Rg (Radius of Gyration) describes the size and shape of the molecule. Smaller Rg values indicate a more compact molecular shape, while larger Rg values indicate a looser molecular shape. Thus, the compactness of protein structures can be assessed through the analysis of Rg plots. As depicted in Fig 7A, the Rg values for the vaccine-HLA-DRB101:01, vaccine-TLR2, and vaccine-TLR4 complexes exhibited an initial decrease followed by fluctuations within a specific range after 20 ns. This indicates that these three complexes became more compact during the simulation and achieved stability at the 20 ns. Meanwhile, though there was no significant decline for the curve of the curve of vaccine-HLA 02:01 complex, it always fluctuates below 4.5 nm, still believing that the complex exists in a relatively tight binding. The average Rgs are 4.02 (vaccine-TLR2), 3.82 (vaccine-TLR4), 4.41 (vaccine- HLA-A02:01) and 4.30 (vaccine- HLA-DRB101:01) (Fig 7B). The number of hydrogen bonds reflects the strength of interactions between protein complexes, more hydrogen bonds indicating stronger interaction between proteins. The hydrogen bonds in the vaccine-HLA-DRB101:01 and vaccine-TLR2 complexes remain stable around 10, the hydrogen bonds in the vaccine-HLA-A02:01 and TLR4 complexes flucture and ultimately stabilize around 20 all complexes were considered stable (Fig 7C). RMSF (Root Mean Square Fluctuation) measures the degree of variation in the positions of atoms within the

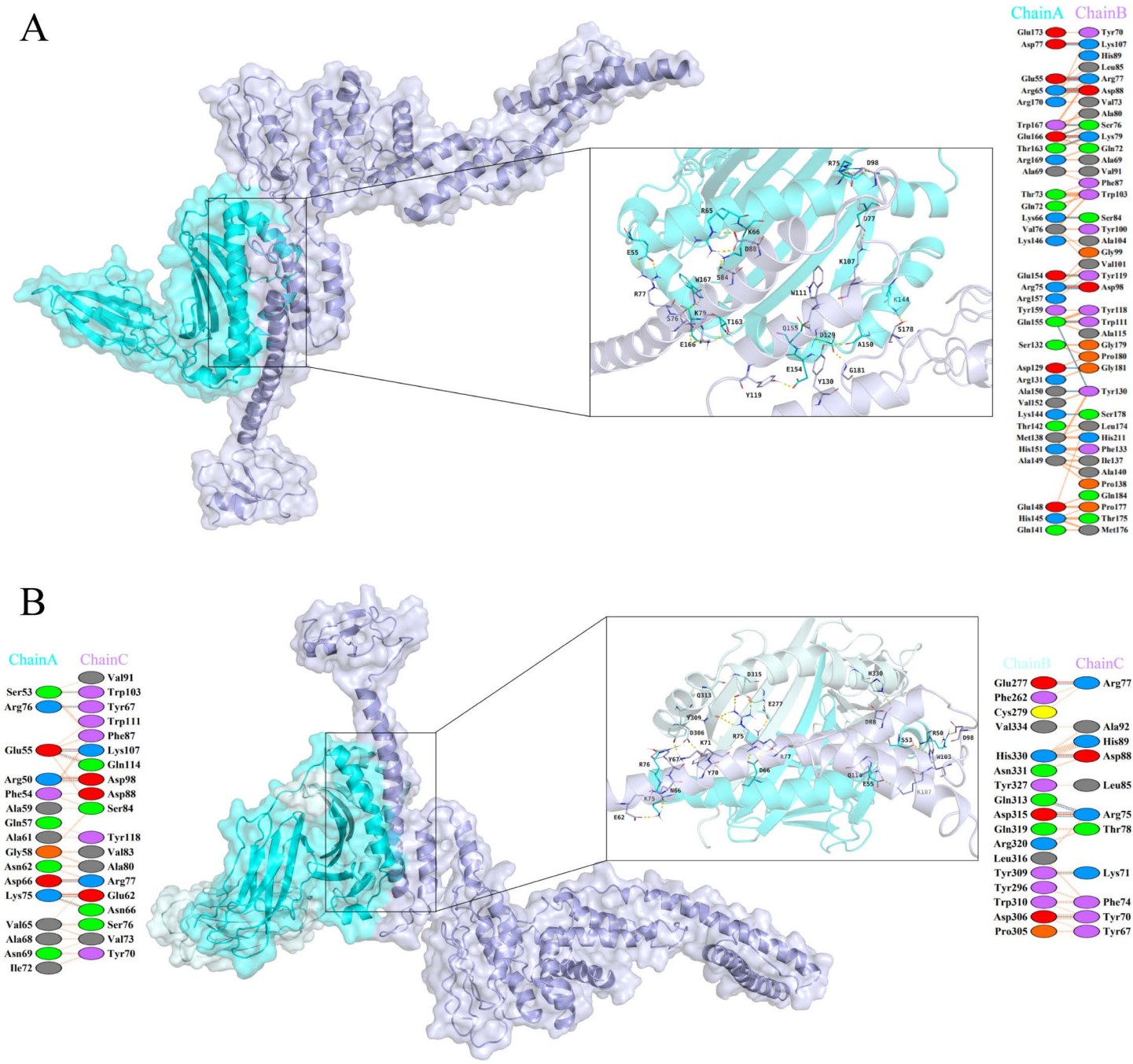

**Fig 5. (A) Docking model of the vaccine with HLA-A02:01 molecule, blue represents the HLA-A02:01 molecule, and purple represents the vaccine molecule; (B) Docking model of the vaccine with HLA-DRB1*01:01 molecule, blue represents the HLA-DRB1*01:01 molecule, and purple represents the vaccine molecule.**

molecule. Smaller RMSF values indicate more stable movements during the simulation process, and in the RMSF analysis of four complexes.

Residues around 1–250 in the vaccine-HLA-A02:01 complex exhibit higher flexibility. Residues in the vaccine-HLA-DRB101:01 complex, overall, exhibit lower flexibility. Residues 380–486, as well as residues 400–486, in the vaccine-TLR2 complex exhibit higher

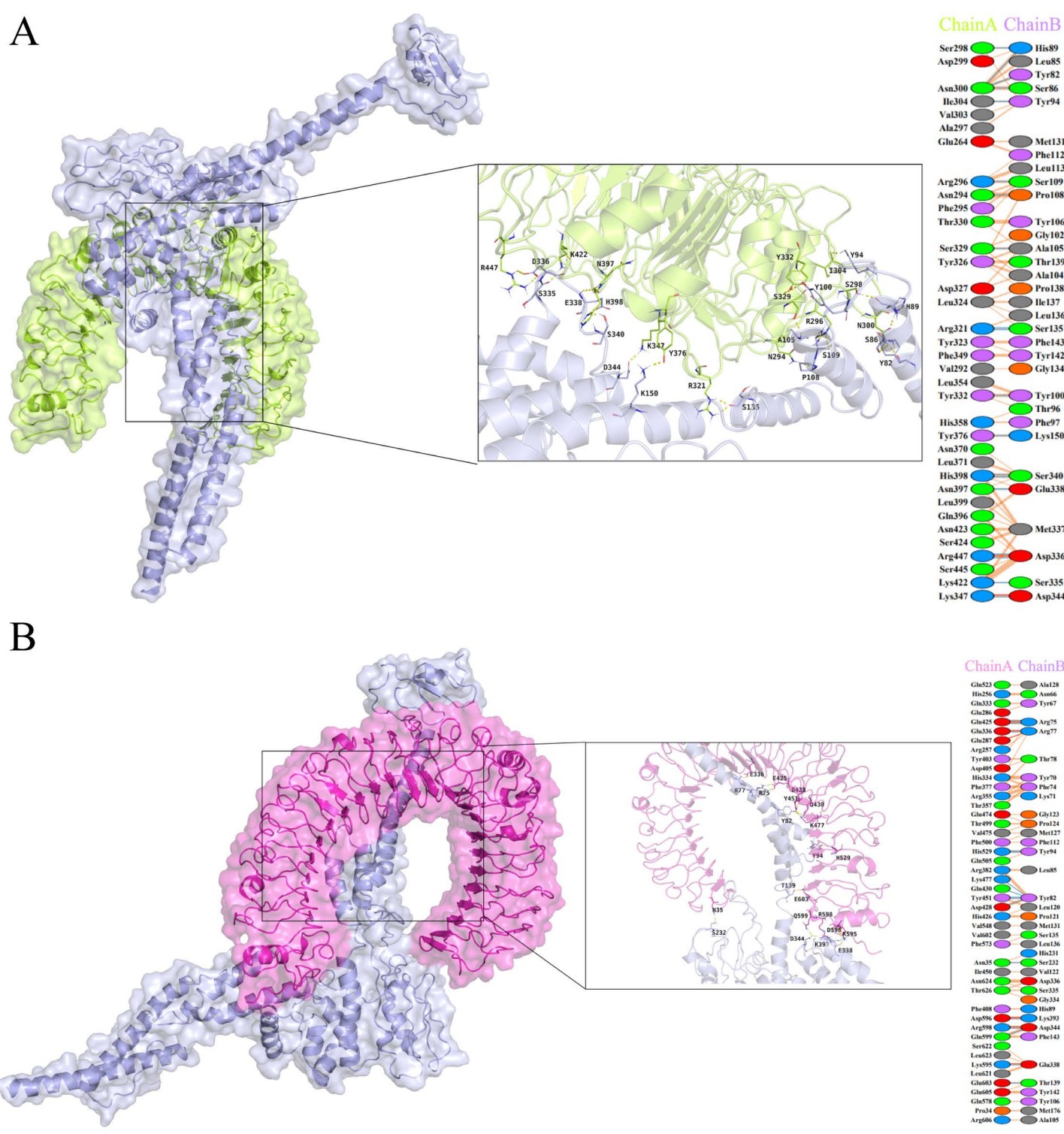

**Fig 6.** **(A) Docking model of the vaccine with TLR2 molecule, green represents the TLR2 molecule, and purple represents the vaccine molecule; (B) Docking model of the vaccine with TLR4 molecule, red represents the TLR4 molecule, and purple represents the vaccine molecule.**

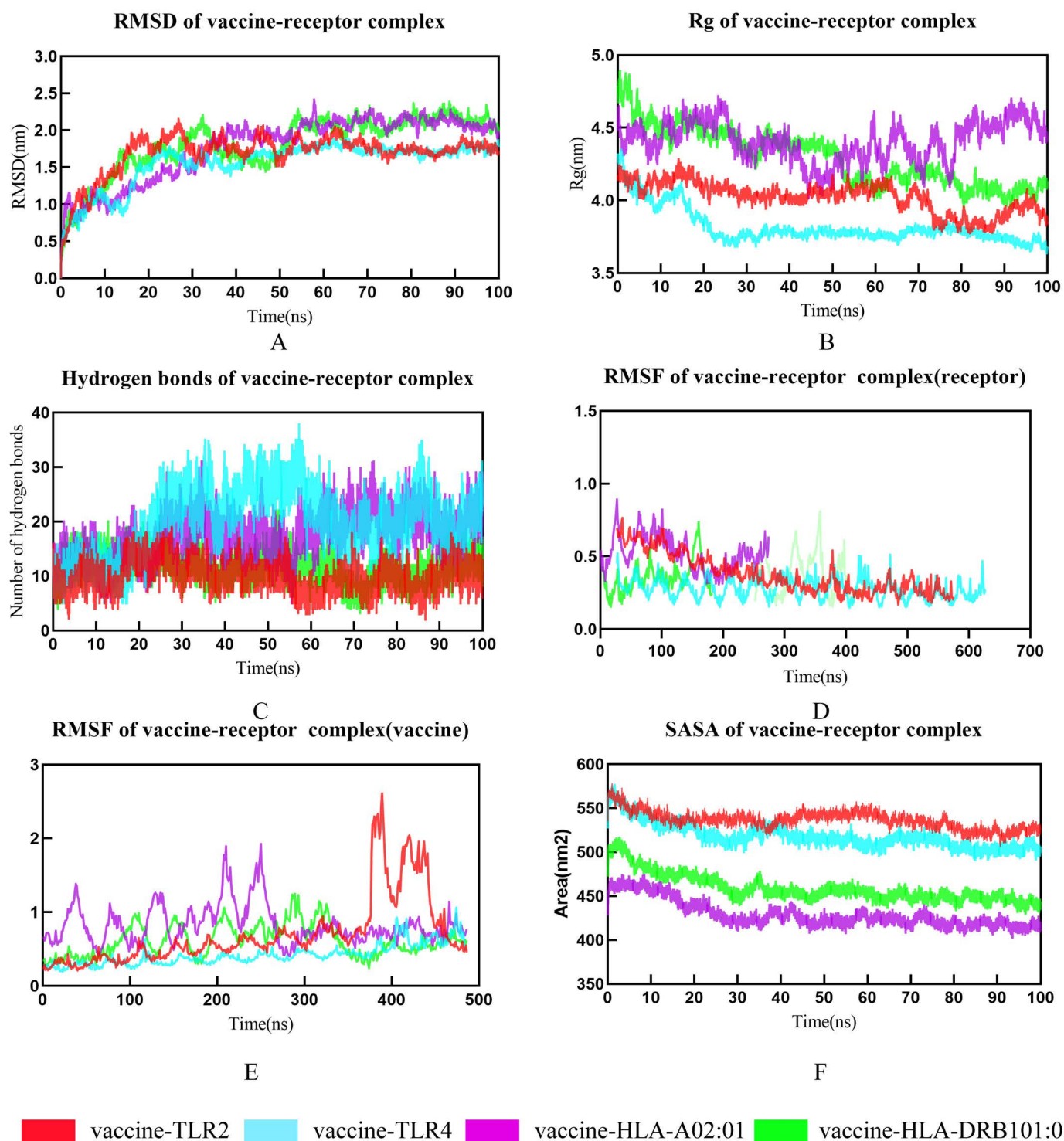

**Fig 7. Molecular dynamic simulation results Blue for vaccine-TLR4 complex, red for vaccine-TLR2 complex, purple for vaccine-HLA-A\*02:01 complex, green for vaccine-HLA-DRB1\*01:01 complex.** (A) Radius of gyration (Rg) plots of vaccine-receptors complexes, suggesting the compactness of complexes; (B) H-bonds formed in complexes; (C, D) RMSF (root mean square fluctuation) of vaccine-receptors, reflects the flexibility and fluctuation of the amino-acids residues in the side chain of docked complexes, receptors' RMSF(C) and vaccines' RMSF(D); (E) Solvent Accessible Surface Area (SASA) of vaccine-receptor complex; (F) RMSD (root mean square deviation) plots of vaccine-receptors, reflects the stability between the vaccine and receptor.

flexibility (Fig 7D and 7E). The SASA (Solvent Accessible Surface Area) refers to the area of the complex molecule that comes into contact with the solvent. Inversely, the smaller the contact area between the complex and the solvent, the larger the contact area between the complex molecules, indicating a tighter binding. SASA of four complexes reduced slightly during the simulation, indicting the binding between vaccine and receptors increased during the 100 ns (Fig 7F).

Further, the GMX_MMPBSA v1.56 calculated the binding energies of the last 20 ns of molecular dynamics simulation trajectories. The calculated binding energies between the vaccine and HLA-A*02:01, HLA-DRB1*01:01, TLR2, and TLR4 were −130.60 KJ/mol, −145.58 KJ/mol, −97.03 KJ/mol, and −199.63 KJ/mol, respectively (Table 6). The result is visualized using the method of residue energy decomposition, where the binding energy of the receptor and ligand is decomposed onto each amino acid residue to identify the residues that contribute to the binding process (Fig 8). These results further demonstrated that the vaccine can effectively interact with immune receptors, and ultimately eliciting host immune responses.

## 7. Population coverage and immunization simulation

IEDB database was utilized for the assessment of population coverage regarding multi-epitope vaccines. The findings revealed that the multi-epitope vaccines provided coverage of 99.32% to the global population, encompassing 99.0% in China, 99.93% in Europe, 100.0% in the United States, and 90.21% in Australia (Fig 9).

The C-IMMSIM server is utilized for simulating the host's immune response to vaccine antigens. During the simulation of the human immune response over an almost one-year period, after three vaccine injections, the antibody levels progressively increase, reaching a peak level approaching 200,000. The antibody levels plateau at 30,000 after 150 days (Fig 10A). Additionally, three vaccine injections can augment the levels of IL2 and IFN-γ in the host (Fig 10B). As depicted in the figures, the population of B, plasma, and helper T cells undergo a sequential increase, followed by a subsequent decrease and eventual equilibrium during three vaccine injections (Fig 10C, 10D and 10E). Activated cytotoxic T cells (TC cells) exhibit an initial increase followed by a decrease, while dormant cytotoxic T cells initially decline and subsequently rise (Fig 10F). MA, DC and NK cells demonstrate an initial upward trend post-vaccination, followed by a subsequent decline, ultimately reaching a relatively stable state (Fig 10G, 10H and 10I). The simulation results suggest that the vaccine can elicit a robust production of antibodies in the host, potentially leading to long-lasting immunity. Nevertheless, additional experimental validation is required to ascertain its protective efficacy.

**Table 6. MMPBSA energy analysis.**

| | Vaccine-HLA-A*02:01 | Vaccine-HLA-DRB1*01:01 | Vaccine-TLR2 | Vaccine-TLR4 |
|---|---|---|---|---|
| Energy Component | | | | |
| ΔVDWAALS | −169.73 (10.68) | −165.48 (9.74) | −122.88 (11.52) | −216.69 (14.51) |
| ΔEEL | −1402.23 (70.99) | −2322.42 (61.16) | −1432.84 (124.51) | −3390.71 (108.47) |
| ΔEPB | 1462.78 (69.00) | 2361.15 (59.56) | 1474.22 (124.66) | 3435.70 (109.79) |
| ΔENPOLAR | −21.42 (1.30) | −18.83 (0.92) | −15.53 (1.27) | −27.93 (1.76) |
| ΔGGAS | −1571.96 (74.09) | −2487.90 (64.67) | −1555.72 (127.62) | −3607.40 (116.69) |
| ΔGSOLV | 1441.36 (68.35) | 2342.32 (59.06) | 1458.69 (124.17) | 3407.78 (108.56) |
| ΔTOTAL | −130.60 (13.42) | −145.58 (12.02) | −97.03 (10.78) | −199.63 (19.95) |

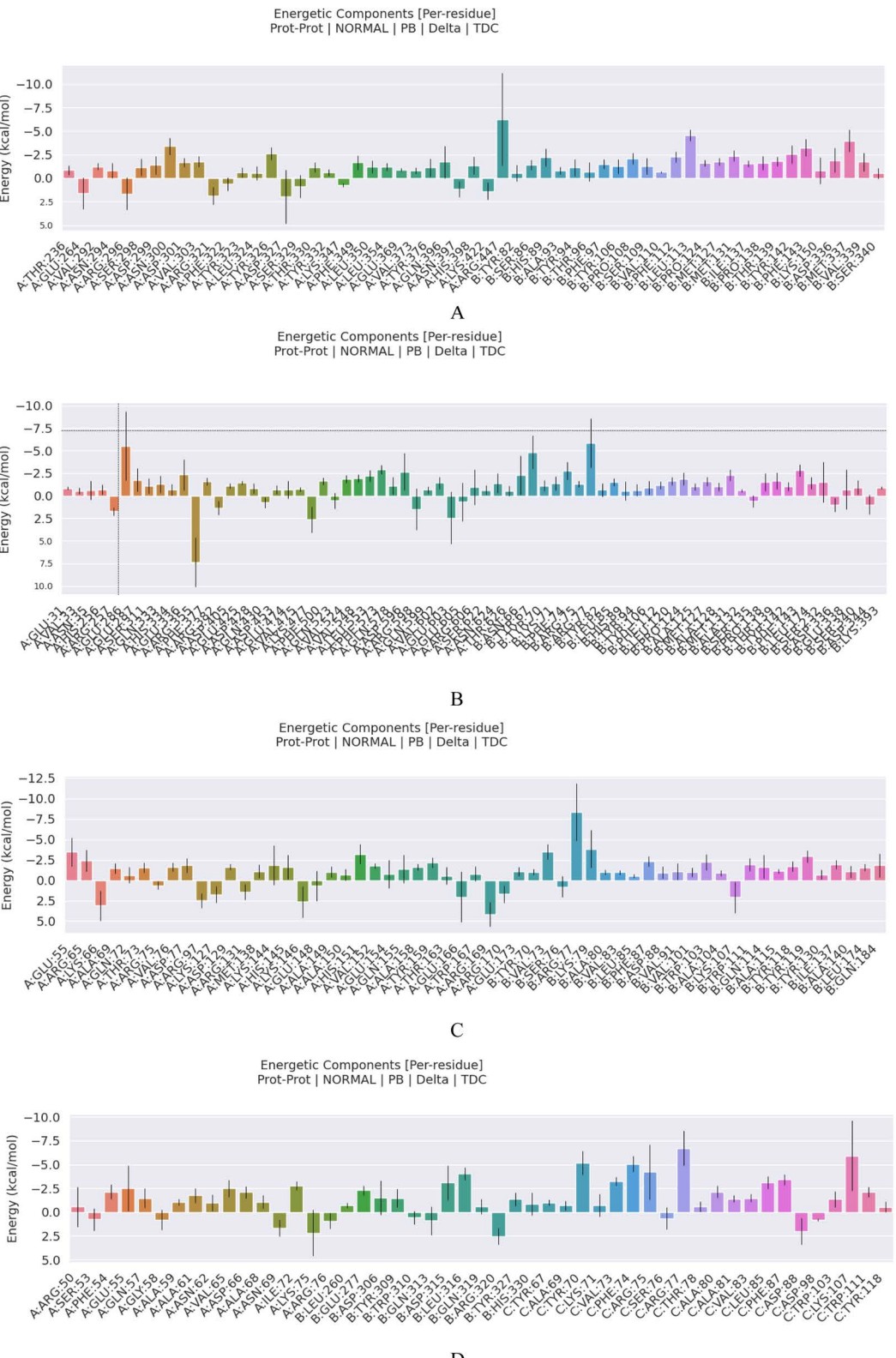

**Fig 8. MMPBSA residue energy decomposition.** (A) Complex of vaccine-TLR2; (B) Complex of vaccine-TLR4; (C) Complex of vaccine-HLA-A*02:01; (D) Complex of vaccine-HLA-DRB1*01:01. In each figure, B molecular stand for the vaccine and A molecular stand for the receptors.

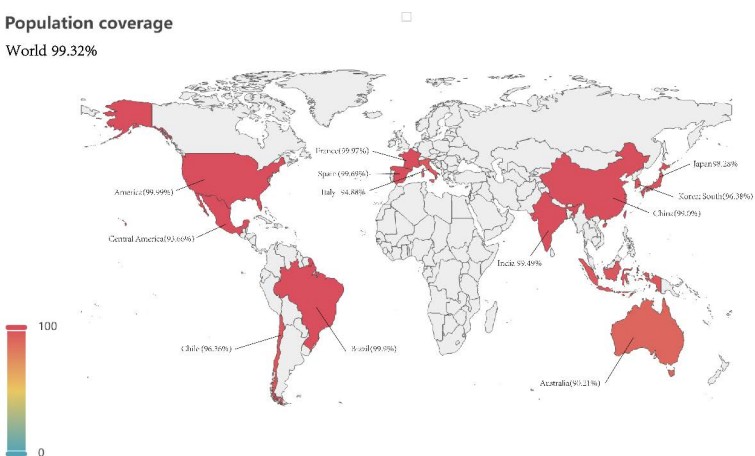

**Fig 9. Population coverage map (based on pyecharts (** https://gitcode.com/gh_mirrors/py/pyecharts/overview?utm_source=highlight_word_gitcode&word=pyecharts**)).**

## 8. Construction of multi-epitope mRNA vaccine and vaccine vector

The Jcat server was utilized for performing reverse translation and optimization. As a result of the improvement process, a 2274 bp cDNA fragment was obtained, containing the full-length epitope vaccine, as well as the tPA and MITD sequences. The cDNA fragment exhibits a CAI value of 0.96 with a 67.28% GC content, both falling within the optimal range, indicating favorable density and thermos-stability. Upon merging with the UTR sequence, the full-length DNA spans 2532 bps. Subsequently, RNAfold analysis the free energy of the thermodynamic ensemble to be −776.90 kcal/mol (S1 and S2 Figs).

Then, the imRNA processing tool did not identify any immunomodulatory motifs in the vaccine mRNA sequence. The RPISeq tool found binding scores of 0.7 (based on the RF method) and 0.62 (based on the SVM method) for the vaccine mRNA with the TLR3 receptor, both scores exceeding 0.5, indicating that the vaccine mRNA may bind to the TLR3 receptor. Besides, the SVM receptor binding scores of TLR7, 8, and 9 were all below 0.5, indicating that they may not be able to bind to the mRNA vaccine (S1 Table). Then, through the HDOCK server, the binding score between the vaccine mRNA and TLR3 receptor was determined to be −491.93, with a confidence score of 0.9989, indicating that the vaccine mRNA can bind to the TLR3 receptor (Fig 11).

Graphical output illustrating these results can be found in the S1 and S2 Figs. The predicted results indicate the stability of the multi-epitope mRNA vaccine. Lastly, the vaccine's DNA sequence was inserted into the Pet28a (+) plasmid between BamHI and XhoI sites for vector construction (S3 Fig).

## Discussion

*SFTSV* is an emerging tick-borne virus that has garnered increasing attention due to its high fatality rate and frequent outbreaks [50]. How to treat and prevent SFTSV has always been the focal point of researchers' attention. Recently, progress has been made in the treatment of *SFTSV*, and calcium channel blockers (CCB) have been found to effectively reduce the mortality rate, CCB can accelerate virus clearance by impairing viral internalization and genomic replication [51]. Prior to this, ribavirin was also considered a potential antiviral

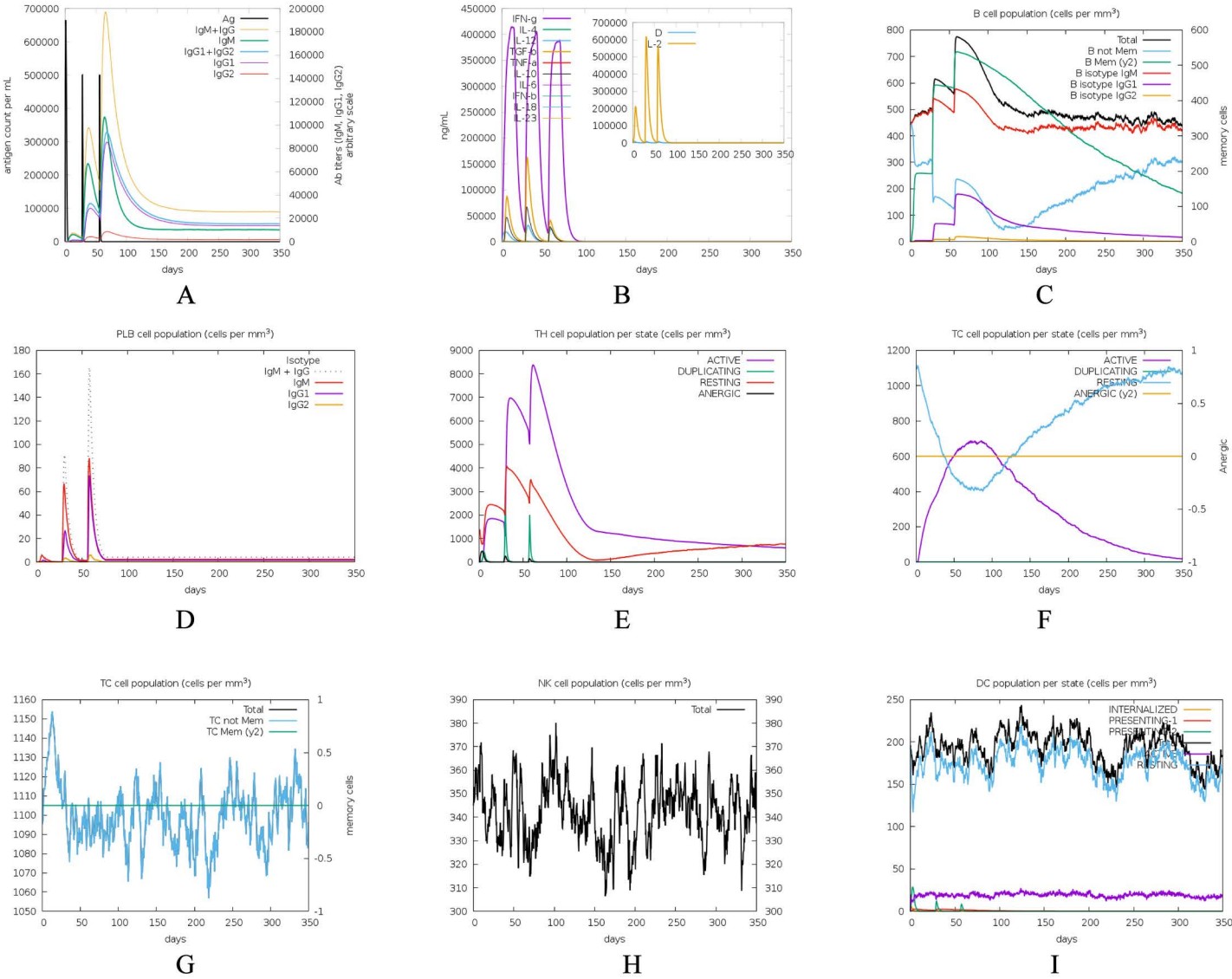

**Fig 10. Immunological simulation analysis.** (A) Antibody levels induced by three doses of vaccine injection; (B) Levels of cytokines such as IL2, IFN-γ induced; (C) Levels of B cells induced; (D) Levels of plasma cells; (E) Levels of helper T (TH) cells induced; (F) Levels of cytotoxic T (TC) cells induced; (G) Levels of MA cells induced; (H) Levels of natural killer (NK) cells induced; (I) Levels of dendritic (DC) cells in different states.

drug [52]. Besides, the gut probiotic *Akkermansia muciniphila* has been found to protect the host from SFTSV infection through its secretions [53]. However, overall, there is still a lack of effective treatment methods for SFTSV. Meanwhile, in the field of vaccine development, there are also some attempts have been made. Live attenuated vaccine platform is a traditional but effective strategy, the existing two types of live attenuated vaccines both achieve reduced viral pathogenicity through the editing of the Ns protein [54], which also indicates that Ns is the main virulence protein of the virus. Compared to live attenuated vaccines, vector vaccines or component vaccines can provide stronger targeting, and VSV-based Gn/Gc vaccine can induce strong humoral immunity [13], which is consistent with our understanding that Gn/Gc can provide potent neutralizing epitopes. The NP or glycoprotein precursor (GPc) vaccine based on the LC16m8 vector can reduce the mortality rate of mice,

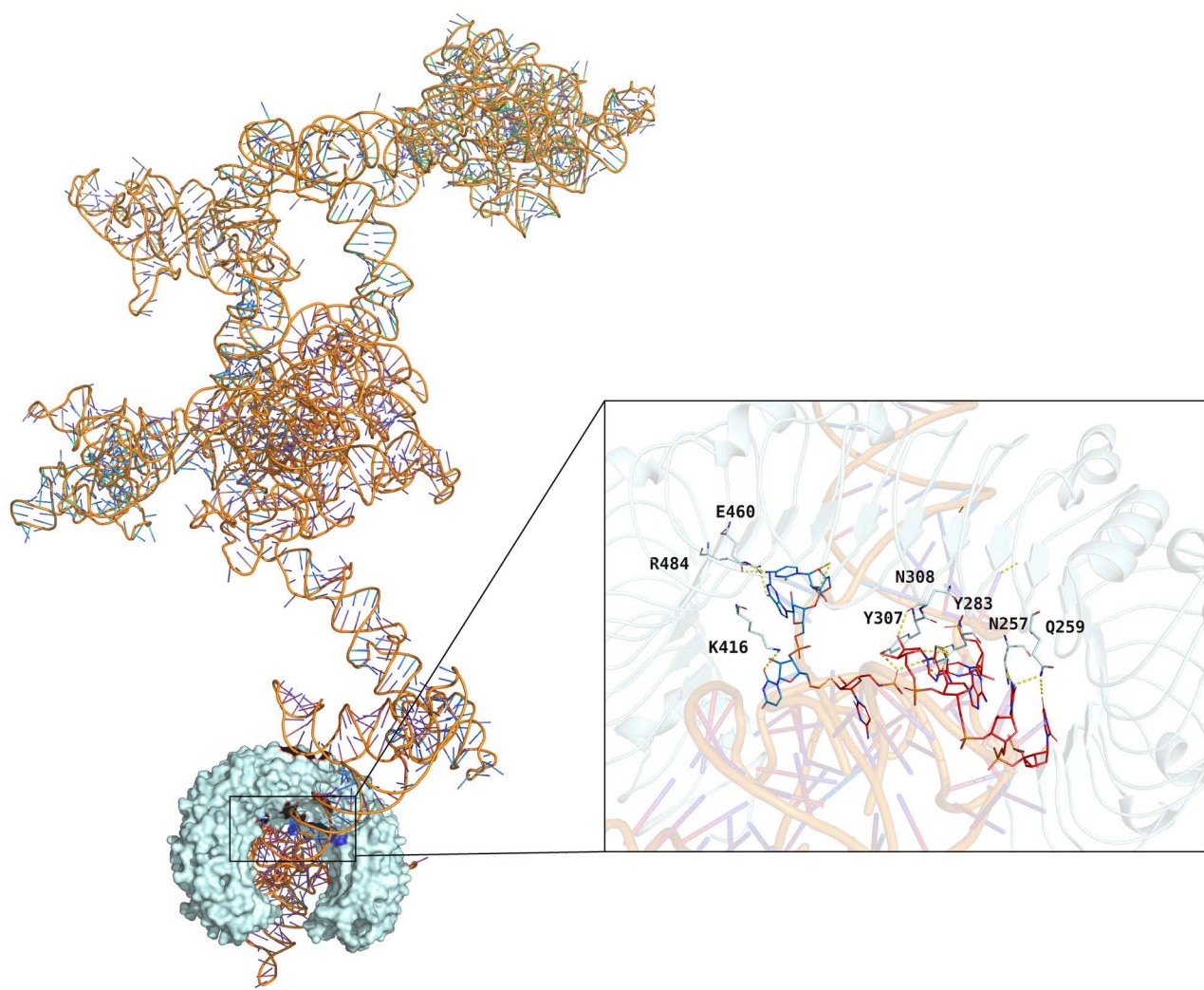

**Fig 11. Docking model of the vaccine (mRNA) with TLR3 molecule.**

and the concurrent humoral transfer experiment once again demonstrates the importance of B-cell immunity in *SFTSV* infection [14]. Nucleic acid vaccines carries antigens coded by DNA or RNA, administering multiple antigens within a single immunization can induce both cellular and humoral immunity, particularly eliciting a strong CD8+ T cell response. Additionally, nucleic acid vaccines are safer and have stronger specificity compared to traditional vaccines [55]. Kang's team developed a novel protective DNA vaccine encoded Gn, Gc, and NP/NS protein, this study also found that stimulating helper T cells with IL-12 can improve the vaccine's efficacy [56]. Compared with DNA vaccine, an mRNA vaccine can provide stronger protection [57], mRNA vaccine can activate a balanced T cell and B cell responses, while reducing the risk of integration into the receptor genome. Furthermore, mRNA vaccines have demonstrated production advantages in fighting against COVID-19, such as flexible antigen design, simple preparation process, and the ability to rapidly scale up production capacity [58].

At this present, our team designed a novel mRNA vaccine and validated the vaccine with in silico methods. The core sequence of the vaccine is conservative epitopes from proteins

Gn, Gc, Np, and NSs, which are considered to generate protective immunity. RNA viruses rely on RdRp for replication and lack proofreading mechanisms, this leads to easy generation of mutations in RNA viruses, allowing them to escape vaccine-induced memory immunity [59]. However, there are always relatively stable core sequences in the viral genome, alignment of 230 known *SFTSV* sequences make sure that multi-epitope vaccine screened from reference sequence of *SFTSV* can cover most strains without losing the protective ability during viral mutations. Meanwhile, to induce both humoral and cellular immune response, 16 T cell epitopes (including 9 CTL epitopes and 7 HTL epitopes) and 8 LBL epitopes were selected for mRNA vaccine construction, in addition to being highly conserved in SFTSV, these epitopes also possess strong antigenicity and suitable immunogenicity. Moreover, no homology were shown between selected epitopes and human or human gut commensal bacteria, to avoid inducing latent autoimmunity damage. To construct a vaccine with epitopes, GPGPG, AAY, and KK linkers connected epitopes, including a PADRE epitope for stimulating additional helper T cells, these linkers are chosen to facilitate cleavage and elicit an enhanced immune response [60,61]. Two adjuvants are connected to the epitope sequence by EAAAK linkers, β-defensin II is attached to the N-terminal to regulate CTL and HTL response, in addition to its well-documented antimicrobial function, β-defensin II plays a crucial role in the chemotaxis of Toll-like receptors (TLRs), particularly TLR2 and TLR4 [62]. TAT sequence is attached to the C-termini for a leading vaccine across the cell membrane and stimulate the phagocytic function of antigen-presenting cells towards the vaccine [63]. Finally, the vaccine contains 486 aas with a molecular weight of 52.08 kDa, for vaccine design and subsequent synthesis, this weight falls within an appropriate range, analysis of the construction's aa composition also indicates that instability index [33.61], and aliphatic [70.39] are within the ideal range. To further validate that the vaccine can synthesize stable proteins and raise immune responses after being taken up by body cells, we predicted the folding conformation of the protein in space with the ab initio prediction method, Ramachandran plot and parameters such as Z-score predict the rationality of protein spatial structures. Given the roles of TLR2 and TLR4 in virus recognition and induction of inflammatory cytokines, we further docked the vaccine protein with both TLR2 and TLR4, meanwhile, docking simulation between vaccine molecules and MHC molecules were also carried out. In all four docking simulations, vaccine molecular docked tightly with receptors. Next, molecular dynamics simulations were used to simulate the interactions inside the vaccine-receptor complexes in a water molecular environment. During the 100 ns computation, the RMSD results indicate the presence of stable interactions between the vaccine and receptor. RMSF (root mean square fluctuation) analysis provided insights into the flexibility of the residues while the radius of gyration (Rg) plots give out the degree of loosening in the complexes, results observed further supported that the complex is stable. In accordance to MD analysis, MM/GBSA estimation validates that the binding affinity between the vaccine and receptors are relatively high.

Furthermore, the detection results of the binding between mRNA and TLR molecules showed that mRNA can bind to TLR3, but may not be able to bind to TLR7, 8, or 9. In virus infection, these TLR molecules may trigger excessive inflammatory responses, even leading to a cytokine storm. However, the binding of mRNA to TLR3 molecules can regulate the activation and proliferation of the TLR signaling pathway, leading to various cascades and cross-signaling pathways, which may help subunit vaccines function more effectively.

At last, C-IMMSIM predicted the immune response generated after vaccine injection, high titer antibody responses were observed after infection, which is very important for the prevention of SFTSV. IgM and IgG increased significantly after vaccination and tended to

maintain a relatively stable titer over a longer period of time, indicating strong and rapid response and long-lasting protective immunity. During the procedure of HTL epitope selection, the potential ability to induce cytokines was an important criterion, and later we noticed the vaccine elicit an increase of IFN-g, IL-2 and IL-4. Meanwhile, powerful cellular and humoral immune responses were observed, vaccination triggers the activation of B and T-cell populations, and ultimately, it forms immunological memory. In general, the vaccine is highly effective as expected.

## Conclusion

In this present, our group have designed a novel mRNA vaccine candidate to combat SFTSV with immunoinformatic methods. We selected multi-epitope mRNA vaccine as a platform. By incorporating immunodominant epitopes, we aimed to achieve improved targeting while minimizing undesirable side effects. Additionally, a design with conserved epitopes can offer a wider range of protection against different subtypes, also, a vaccine containing T and B cell epitopes are expected to elicit both humoral and cellular immune responses. Further experimental verification is required for this mRNA vaccine in vivo and vitro. After the procedure of transcription, capping, and packaging, the vaccine will be injected into mice to observe the immune response. It should be pointed out that the final efficacy of the vaccine still needs to be determined through in vitro and in vivo experiments. But in summary, the design of this vaccine is a promising vaccine candidate that performs well in simulations. Given the limited research on SFTSV vaccines, it can provide an effective strategy for the prevention of SFTSV.

## Supporting information

**S1 Text. Multi-epitope vaccine core sequence, core multi-epitope vaccine sequences containing HTL, CTL, and linear B-cell epitopes, as well as linkers between adjuvants and components.**
(DOCX)

**S2 Text. tPA sequence, MITD sequence, Kozak sequence, several auxiliary component original sequences used to compose the final vaccine sequence.**
(DOCX)

**S3 Text. 5'UTR and 3'UTR, The 5'UTR and 3'UTR sequence used for RNA vaccine sequence construction.**
(DOCX)

**S4 Text. Core-cDNA, cDNA sequence corresponding to the core vaccine sequence.**
(DOCX)

**S5 Text. RNA, Target RNA sequence of the vaccine.**
(DOCX)

**S6 Text. Full-cDNA, the complete cDNA sequence containing 5' UTR, 3' UTR, vaccine sequence, and all auxiliary sequences, directly used for synthesizing RNA vaccine sequence.**
(DOCX)

**S1 Fig. MFE secondary structure and MFE Centroid structure of vaccine mRNA sequence.**
(TIF)

**S2 Fig. MFE Centroid structure of vaccine mRNA sequence.**
(TIF)

**S3 Fig. Plasmid construction, schematic diagram of the final constructed plasmid structure.**
(TIF)

**S1 Table. Prediction of mRNA binding parameters with TLRs.**
(DOCX)

**S1 Data. Validation of epitopes with new server.**
(ZIP)

**S2 Data. Homology testing of intestinal microbiota.**
(ZIP)

## Author contributions

**Conceptualization:** Fei Zhu, Shiyang Ma, Jie Chen.

**Formal analysis:** Yizhong Xu, Ziyou Zhou.

**Funding acquisition:** Pinhua Pan.

**Methodology:** Fei Zhu, Shiyang Ma, Hang Yang, Jie Chen.

**Project administration:** Pinhua Pan.

**Software:** Fei Zhu, Shiyang Ma, Peipei Zhang, Hang Yang, Caixia Tan.

**Supervision:** Pinhua Pan.

**Validation:** Shiyang Ma, Yizhong Xu, Ziyou Zhou, Wenzhong Peng.

**Visualization:** Jie Chen.

**Writing – original draft:** Shiyang Ma, Ziyou Zhou, Peipei Zhang, Caixia Tan.

**Writing – review & editing:** Jie Chen.

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
