## [Decision Letter · Decision Letter 0]

12 Aug 2024

Dear Dr. pan,

Thank you very much for submitting your manuscript "Development of a novel multi-epitope mRNA vaccine candidate to combat SFTSV pandemic" for consideration at PLOS Neglected Tropical Diseases. As with all papers reviewed by the journal, your manuscript was reviewed by members of the editorial board and by several independent reviewers. In light of the reviews (below this email), we would like to invite the resubmission of a significantly-revised version that takes into account the reviewers' comments. 

We cannot make any decision about publication until we have seen the revised manuscript and your response to the reviewers' comments. Your revised manuscript is also likely to be sent to reviewers for further evaluation.

Sincerely,

Jun Jiao

Academic Editor

Elvina Viennet

Section Editor

Reviewer's Responses to Questions

**Key Review Criteria Required for Acceptance?**

**Methods**

-Are the objectives of the study clearly articulated with a clear testable hypothesis stated?

-Is the study design appropriate to address the stated objectives?

-Is the population clearly described and appropriate for the hypothesis being tested?

-Is the sample size sufficient to ensure adequate power to address the hypothesis being tested?

-Were correct statistical analysis used to support conclusions?

-Are there concerns about ethical or regulatory requirements being met?

Reviewer #1: 1. Yes, objectives of the study is clearly articulated with a clear testable hypothesis stated

2. Yes, study design is appropriate to address the stated objectives.

3. Authors have not mentioned anything regarding the population or sample size.

4. Yes, correct statistical analysis used to support conclusions.

5. there are no concerns about ethical or regulatory requirements.

Reviewer #2: - Are the objectives of the study clearly articulated with a clear testable hypothesis stated?

- The objectives to develop a multi-epitope mRNA vaccine for SFTSV and validate it with in silico methods are clearly articulated. 

- Is the study design appropriate to address the stated objectives?

- The study design involving epitope prediction, vaccine construction, and validation through molecular docking and dynamics simulations is appropriate and comprehensive for addressing the development and preliminary evaluation of a vaccine candidate.

- Is the population clearly described and appropriate for the hypothesis being tested?

- The authors utilized IEDB tools to estimate the population coverage of the novel vaccine candidate. Their findings indicate that the multi-epitope vaccines achieve a remarkable coverage rate, reaching 99.32% of the global population. Specifically, the coverage includes 99.0% in China, 99.93% in Europe, 100.0% in the United States, and 90.21% in Australia.

- Is the sample size sufficient to ensure adequate power to address the hypothesis being tested?

- In the context of this in silico study, the 'sample size' can be considered as the number of epitopes and simulations run. The study appears to have a comprehensive set of epitope predictions and simulations, which seems sufficient for initial validation.

- Were correct statistical analysis used to support conclusions?

- The paper does not specify statistical methods typically because it primarily focuses on in silico predictions and simulations. 

- Are there concerns about ethical or regulatory requirements being met?

- As an in silico study, typical ethical concerns applicable to clinical or in vivo studies do not apply here.

Reviewer #3: The objectives of the study were clearly articulated and the methods used to achieve this were well described. I appreciate the detail of the resulting final construct structures and how they were optimized. However, I did not see a comparison between the structure of each individual epitope and the corresponding structure of that epitope in the SFTSV gene. The vaccine was optimized immunogenicity, but the structural relationship between the vaccine and the parent virus must be demonstrated.

Also, I wonder about the choice for exploring the recognition by TLR 2 and TLR 4 based on the rationale of "given the roles of TLR2 and TLR4 in virus recognition..." What is the expected benefit to recognition of the vaccine by these TLRs, and why not evaluate the TLRs more strongly associated with virus recognition, like TLR 3 and 7, which recognize viral RNA during replication?

**Results**

-Does the analysis presented match the analysis plan?

-Are the results clearly and completely presented?

-Are the figures (Tables, Images) of sufficient quality for clarity?

Reviewer #1: Yes, all those analysis presented here is match the analysis plan

Yes, results are clearly and completely presented.

Yes, figures are (Tables, Images) sufficient quality for clarity.

Reviewer #2: The computing results presented align with the methods described, including epitope prediction, vaccine construction, and simulation results.

Reviewer #3: The results are clear overall, and the figures are of excellent quality. 

I do have questions about the result described from line 358-371. The antibodies were predicted to reach a peak level of 200,000, however the units for this are described as arbitrary. How might this antibody level relate to binding and/or neutralization titers typically described from in vivo studies? Also, there is a list of changes presented here in the levels of various cell types and cytokines, but no description of what the implication of these changes would be with respect to immunogenicity and/or vaccine protection, nor are they related to the summary statement of that paragraph describing the simulation results as antibody production.

**Conclusions**

-Are the conclusions supported by the data presented?

-Are the limitations of analysis clearly described?

-Do the authors discuss how these data can be helpful to advance our understanding of the topic under study?

-Is public health relevance addressed?

Reviewer #1: 1.Yes, the conclusions are supported by the data presented.

2.No, the authors have not described some limitations associated with using several bioinformatics tools, software, or servers. If there are any, the authors should inform the readers.

3.Yes, the authors have discussed very well how these data can help advance our understanding of the topic under study.

Reviewer #2: - Are the conclusions supported by the data presented?

- The potential efficacy of the vaccine is supported by in silico studies.

- Are the limitations of analysis clearly described?

- The manuscript acknowledges the limitations inherent in in silico studies and the need for further in vivo and in vitro testing.

- Do the authors discuss how these data can be helpful to advance our understanding of the topic under study?

- The study discusses the implications of the findings for future vaccine development against SFTSV, which is valuable for the field.

- Is public health relevance addressed?

- The manuscript highlights the potential public health impact of an effective vaccine against SFTSV, especially considering the virus's high fatality rate and pandemic potential as mentioned in the introduction.

Reviewer #3: (No Response)

**Editorial and Data Presentation Modifications?**

Reviewer #1: Minor Revision

Reviewer #2: Minor

1. The caption of Table 4, currently labeled as 'formation B epitopes', should be revised to 'conformational B-cell epitopes' for clarity and accuracy.

2. In the 'Epitope screening' part of the abstract figure, it appears that the authors use MHC-peptide binding process to represent the epitope. However, CTL should correspond to MHCI, which is depicted with a single leg (alpha3), and HTL should correspond to MHCII, which is depicted with two legs (alpha2 and beta2). Therefore, the two diagrams should be swapped for accuracy.

3. Line 291: 'epitopes in the structure(Table 4), the overall score varied from '0.855-0.528'. Please consider revise it to 'The overall score varied from 0.528 to 0.855.'

Reviewer #3: The manuscript would benefit from further language refinement, and there are a few typos. Also, the acronyms "HTL" and "LBL" do not seem to exist outside of this manuscript and they feel strange. LBL in particular does not make sense in context (CTL and HTL refer to cell types, whereas there is no such cell type as a "Linear B cell"). Statements regarding LBL, especially in the Abstract, should be clarified to refer to "B cell epitopes that are linear" instead of the implied "epitopes recognized by Linear B cells").

**Summary and General Comments**

Reviewer #1: (No Response)

Reviewer #2: This study presents a thorough in silico investigation into the development of a novel mRNA vaccine against SFTSV. The manuscript offers a detailed and clear description of the construction of a vaccine candidate and its validation using in silico methods. The procedure is both comprehensive and innovative. This research has the potential to significantly accelerate the development of mRNA vaccines against SFTSV. Overall, this paper provides valuable insights and is well-written.

However, several concerns are inevitably still identified and should be addressed before considering this study for publication in this journal.

Major:

1. Some of the bioinformatics tools employed in this study are outdated, potentially compromising the accuracy of the results. Specifically, for predicting CTL epitopes, the authors utilized NetCTL 1.2, a method introduced in 2007. To enhance accuracy and reliability, it is recommended to use more advanced tools like NetMHCpan-4.1 or MixMHCpred-3.0. The authors should consider validating their proposed vaccine using these state-of-the-art tools to substantiate their findings.

2. There is insufficient evidence to validate the claim regarding autoimmunity. In the discussion, the authors stated, "Moreover, no homology was shown between selected epitopes and human or human gut commensal bacteria, to avoid inducing latent autoimmunity damage." However, the tools used to calculate this homology were not specified. Additionally, the threshold for identity that distinguishes homology from non-homology was not mentioned. The authors are encouraged to provide further details on the methodologies and criteria used for these homology assessments.

Minor

1. The caption of Table 4, currently labeled as 'formation B epitopes', should be revised to 'conformational B-cell epitopes' for clarity and accuracy.

2. In the 'Epitope screening' part of the abstract figure, it appears that the authors use MHC-peptide binding process to represent the epitopes. However, CTL should correspond to MHCI, which is depicted with a single leg (alpha3), and HTL should correspond to MHCII, which is depicted with two legs (alpha2 and beta2). Therefore, the two diagrams should be swapped for accuracy.

3. Line 291: 'epitopes in the structure(Table 4), the overall score varied from '0.855-0.528'. Please consider revise it to 'The overall score varied from 0.528 to 0.855.'

Reviewer #3: (No Response)

PLOS authors have the option to publish the peer review history of their article (what does this mean? ). If published, this will include your full peer review and any attached files.

**Do you want your identity to be public for this peer review?** For information about this choice, including consent withdrawal, please see our Privacy Policy .

Reviewer #1: Yes: Swarnendu Basak

Reviewer #2: No

Reviewer #3: No

Figure Files:

Data Requirements:

Please note that, as a condition of publication, PLOS' data policy requires that you make available all data used to draw the conclusions outlined in your manuscript. Data must be deposited in an appropriate repository, included within the body of the manuscript, or uploaded as supporting information. This includes all numerical values that were used to generate graphs, histograms etc.. For an example see here: http://www.plosbiology.org/article/info%3Adoi%2F10.1371%2Fjournal.pbio.1001908#s5 .
---

## [Decision Letter · Decision Letter 1]

29 Dec 2024

Dear Dr. pan,

We are pleased to inform you that your manuscript 'Development of a novel multi-epitope mRNA vaccine candidate to combat SFTSV pandemic' has been provisionally accepted for publication in PLOS Neglected Tropical Diseases.

Best regards,

Jun Jiao

Academic Editor

Elvina Viennet

Section Editor

Shaden Kamhawi

co-Editor-in-Chief

Paul Brindley

co-Editor-in-Chief

Reviewer's Responses to Questions

**Key Review Criteria Required for Acceptance?**

**Methods**

-Are the objectives of the study clearly articulated with a clear testable hypothesis stated?

-Is the study design appropriate to address the stated objectives?

-Is the population clearly described and appropriate for the hypothesis being tested?

-Is the sample size sufficient to ensure adequate power to address the hypothesis being tested?

-Were correct statistical analysis used to support conclusions?

-Are there concerns about ethical or regulatory requirements being met?

Reviewer #1: Yes objectives of the study clearly articulated with a clear testable hypothesis stated.

Yes study design appropriate to address the stated objectives.

its an Insilco vaccine efficacy study, so not applicable for the information regarding population and sample size.

Yes correct statistical analysis used to support conclusions.

No concerns about ethical or regulatory requirements being met.

Reviewer #2: (No Response)

Reviewer #4: .

**Results**

-Does the analysis presented match the analysis plan?

-Are the results clearly and completely presented?

-Are the figures (Tables, Images) of sufficient quality for clarity?

Reviewer #1: yes the analysis presented match the analysis plan.

yes the results clearly and completely presented.

yes the figures (Tables, Images) of sufficient quality for clarity.

Reviewer #2: (No Response)

Reviewer #4: .

**Conclusions**

-Are the conclusions supported by the data presented?

-Are the limitations of analysis clearly described?

-Do the authors discuss how these data can be helpful to advance our understanding of the topic under study?

-Is public health relevance addressed?

Reviewer #1: yes conclusions supported by the data presented.

yes limitations of analysis clearly described.

yes authors discuss how these data can be helpful to advance our understanding of the topic under study.

Reviewer #2: (No Response)

Reviewer #4: .

**Editorial and Data Presentation Modifications?**

Reviewer #1: Accept

Reviewer #2: (No Response)

Reviewer #4: .

**Summary and General Comments**

Reviewer #1: The detailed screening process to identify the epitope regions is well-designed and impressive. Authors have screened and verified the allergenicity and toxicity of the epitopes, which is important and necessary for studying in silico vaccine safety assessment.

Reviewer #2: The authors have addressed all of my comments in a very detailed and extensive revision. I do not have further comments.

Reviewer #4: When I read this title, I thought that the vaccine developed by the author would be effective. However, after reading the full text, I found that the study was a dry experiment and there was no vaccine effectiveness evaluation experiment, which was very disappointing to me. I think this is at most a vaccine design study, so I think the author's title is exaggerated.

In order to match the title of the paper, I think the authors should add the vaccine effectiveness evaluation experiment, so as to be meaningful. So I gave an major revision.

Of course, authors also have the right to modify their titles to correspond to their actual research.

Overall, the authors conduct a large number of bioinformatic analyses, which can inspire readers to follow this process for vaccine design for a wide variety of other pathogens. From this point of view, I think the significance of this paper is better.

PLOS authors have the option to publish the peer review history of their article (what does this mean? ). If published, this will include your full peer review and any attached files.

**Do you want your identity to be public for this peer review?** For information about this choice, including consent withdrawal, please see our Privacy Policy .

Reviewer #1: **Yes: ** Swarnendu Basak

Reviewer #2: No

Reviewer #4: No

---

## [Editor Report · Acceptance letter]

Dear Dr. pan,

We are delighted to inform you that your manuscript, "Development of a novel multi-epitope mRNA vaccine candidate to combat SFTSV pandemic," has been formally accepted for publication in PLOS Neglected Tropical Diseases.

Best regards,

Shaden Kamhawi

co-Editor-in-Chief

Paul Brindley

co-Editor-in-Chief
